# TOP-LABEL CALIBRATION
# AND MULTICLASS-TO-BINARY REDUCTIONS

**Chirag Gupta & Aaditya Ramdas**
Carnegie Mellon University
{chiragg,aramdas}@cmu.edu

## ABSTRACT

We propose a new notion of multiclass calibration called top-label calibration. A classifier is said to be top-label calibrated if the reported probability for the predicted class label—the top-label—is calibrated, conditioned on the top-label. This conditioning is essential for practical utility of the calibration property, since the top-label is always reported and we must condition on what is reported. However, the popular notion of confidence calibration erroneously skips this conditioning. Furthermore, we outline a multiclass-to-binary (M2B) reduction framework that unifies confidence, top-label, and class-wise calibration, among others. As its name suggests, M2B works by reducing multiclass calibration to different binary calibration problems; various types of multiclass calibration can then be achieved using simple binary calibration routines. We instantiate the M2B framework with the well-studied histogram binning (HB) binary calibrator, and prove that the overall procedure is multiclass calibrated without making any assumptions on the underlying data distribution. In an empirical evaluation with four deep net architectures on CIFAR-10 and CIFAR-100, we find that the M2B + HB procedure achieves lower top-label and class-wise calibration error than other approaches such as temperature scaling. Code for this work is available at https://github.com/aigen/df-posthoc-calibration.

## 1 INTRODUCTION

Machine learning models often make probabilistic predictions. The ideal prediction is the true conditional distribution of the output given the input. However, nature never reveals true probability distributions, making it infeasible to achieve this ideal in most situations. Instead, there is significant interest towards designing models that are calibrated, which is often feasible. We motivate the definition of calibration using a standard example of predicting the probability of rain. Suppose a meteorologist claims that the probability of rain on a particular day is $0.7$. Regardless of whether it rains on that day or not, we cannot know if $0.7$ was the *underlying probability of rain*. However, we can test if the meteorologist is calibrated in the long run, by checking if on the D days when $0.7$ was predicted, it indeed rained on around $0.7D$ days (and the same is true for other probabilities).

This example is readily converted to a formal binary calibration setting. Denote a random (feature, label)-pair as $(X, Y) \in \mathcal{X} \times \{0, 1\}$, where $\mathcal{X}$ is the feature space. A probabilistic predictor $h : \mathcal{X} \to [0, 1]$ is said to be calibrated if for every prediction $q \in [0, 1]$, $\Pr(Y = 1 \mid h(X) = q) = q$ (almost surely). Arguably, if an ML classification model produces such calibrated scores for the classes, downstream users of the model can reliably use its predictions for a broader set of tasks.

Our focus in this paper is calibration for multiclass classification, with $L \geqslant 3$ classes and $Y \in [L] := \{1, 2, \ldots, L \geqslant 3\}$. We assume all (training and test) data is drawn i.i.d. from a fixed distribution $P$, and denote a general point from this distribution as $(X, Y) \sim P$. Consider a typical multiclass predictor, $\mathbf{h} : \mathcal{X} \to \Delta^{L-1}$, whose range $\Delta^{L-1}$ is the probability simplex in $\mathbb{R}^L$. A natural notion of calibration for $\mathbf{h}$, called *canonical calibration* is the following: for every $l \in [L]$, $P(Y = l \mid \mathbf{h}(X) = \mathbf{q}) = q_l$ ($q_l$ denotes the $l$-th component of $\mathbf{q}$). However, canonical calibration becomes infeasible to achieve or verify once $L$ is even $4$ or $5$ (Vaicenavicius et al., 2019). Thus, there is interest in studying statistically feasible relaxations of canonical notion, such as confidence calibration (Guo et al., 2017) and class-wise calibration (Kull et al., 2017).

In particular, the notion of confidence calibration (Guo et al., 2017) has been popular recently. A model is confidence calibrated if the following is true: "when the reported confidence for the predicted class is $q \in [0, 1]$, the accuracy is also $q$". In any practical setting, the confidence $q$ is never reported alone; it is always reported along with the actual class prediction $l \in [L]$. One may expect that if a model is confidence calibrated, the following also holds: "when the class $l$ is predicted with confidence $q$, the probability of the actual class being $l$ is also $q$"? Unfortunately, this expectation is rarely met—there exist confidence calibrated classifier for whom the latter statement is grossly violated for all classes (Example 1). On the other hand, our proposed notion of top-label calibration enforces the latter statement. It is philosophically more coherent, because it requires conditioning on all relevant reported quantities (both the predicted top label and our confidence in it). In Section 2, we argue further that top-label calibration is a simple and practically meaningful replacement of confidence calibration.

In Section 3, we unify top-label, confidence, and a number of other popular notions of multiclass calibration into the framework of multiclass-to-binary (M2B) reductions. The M2B framework relies on the simple observation that each of these notions internally verifies binary calibration claims. As a consequence, each M2B notion of calibration can be achieved by solving a number of binary calibration problems. With the M2B framework at our disposal, all of the rich literature on binary calibration can now be used for multiclass calibration. We illustrate this by instantiating the M2B framework with the binary calibration algorithm of histogram binning or HB (Zadrozny and Elkan, 2001; Gupta and Ramdas, 2021). The M2B + HB procedure achieves state-of-the-art results with respect to standard notions of calibration error (Section 4). Further, we show that our procedure is provably calibrated for arbitrary data-generating distributions. The formal theorems are delayed to Appendices B, C (due to space limitations), but an informal result is presented in Section 4.

## 2 MODIFYING CONFIDENCE CALIBRATION TO TOP-LABEL CALIBRATION

Let $c : \mathcal{X} \to [L]$ denote a classifier or top-label predictor and $h : \mathcal{X} \to [0, 1]$ a function that provides a confidence or probability score for the top-label $c(X)$. The predictor $(c, h)$ is said to be confidence calibrated (for the data-generating distribution $P$) if

$$P(Y = c(X) \mid h(X)) = h(X). \tag{1}$$

In other words, when the reported confidence $h(X)$ equals $p \in [0, 1]$, then the fraction of instances where the predicted label is correct also approximately equals $p$. Note that for an $L$-dimensional predictor $\mathbf{h} : \mathcal{X} \to \Delta^{L-1}$, one would use $c(\cdot) = \arg\max_{l \in [L]} h_l(\cdot)$ and $h(\cdot) = h_{c(\cdot)}(\cdot)$; ties are broken arbitrarily. Then $\mathbf{h}$ is confidence calibrated if the corresponding $(c, h)$ satisfies (1).

Confidence calibration is most applicable in high-accuracy settings where we trust the label prediction $c(x)$. For instance, if a high-accuracy cancer-grade-prediction model predicts a patient as having "95% grade III, 3% grade II, and 2% grade I", we would suggest the patient to undergo an invasive treatment. However, we may want to know (and control) the number of non-grade-III patients that were given this suggestion incorrectly. In other words, is Pr(cancer is not grade III | cancer is predicted to be of grade III with confidence 95%) equal to 5%? It would appear that by focusing on the the probability of the predicted label, confidence calibration enforces such control.

However, as we illustrate next, confidence calibration fails at this goal by providing a guarantee that is neither practically interpretable, nor actionable. Translating the probabilistic statement (1) into words, we ascertain that confidence calibration leads to guarantees of the form: "if the confidence $h(X)$ in the top-label is 0.6, then the accuracy (frequency with which $Y$ equals $c(X)$) is 0.6". Such a guarantee is not very useful. Suppose a patient P is informed (based on their symptoms $X$), that they are most likely to have a certain disease D with probability 0.6. Further patient P is told that this score is confidence calibrated. P can now infer the following: "among all patients who have probability 0.6 of having *some unspecified* disease, the fraction who have *that unspecified* disease is also 0.6." However, P is concerned only about disease D, and not about other diseases. That is, P wants to know the probability of having D *among patients who were predicted to have disease D with confidence 0.6*, not among patients who were predicted to have *some* disease with confidence 0.6. In other words, P cares about the occurrence of D among patients who were told the same thing that P has been told. It is tempting to wish that the confidence calibrated probability 0.6 has any bearing on what P cares about. However, this faith is misguided, as the above reasoning suggests, and further illustrated through the following example.

**Example 1.** Suppose the instance space is $(X, Y) \in \{a, b\} \times \{1, 2, \ldots\}$. ($X$ can be seen as the random patient, and $Y$ as the disease they are suffering from.) Consider a predictor $(c, h)$ and let the values taken by $(X, Y, c, h)$ be as follows:

| Feature $x$ | $P(X = x)$ | Class prediction $c(x)$ | Confidence $h(x)$ | $P(Y = c(X) \mid X = x)$ |
|:---:|:---:|:---:|:---:|:---:|
| $a$ | 0.5 | 1 | 0.6 | 0.2 |
| $b$ | 0.5 | 2 | 0.6 | 1.0 |

The table specifies only the probabilities $P(Y = c(X) \mid X = x)$; the probabilities $P(Y = l \mid X = x)$, $l \neq c(x)$, can be set arbitrarily. We verify that $(c, h)$ is confidence calibrated: $P(Y = c(X) \mid h(X) = 0.6) = 0.5(P(Y = 1 \mid X = a) + P(Y = 2 \mid X = b)) = 0.5(0.2 + 1) = 0.6$. However, whether the actual instance is $X = a$ or $X = b$, *the probabilistic claim of* 0.6 *bears no correspondence with reality*. If $X = a$, $h(X) = 0.6$ is extremely overconfident since $P(Y = 1 \mid X = a) = 0.2$. Contrarily, if $X = b$, $h(X) = 0.6$ is extremely underconfident. $\square$

The reason for the strange behavior above is that the probability $P(Y = c(X) \mid h(X))$ is not interpretable from a decision-making perspective. In practice, we never report just the confidence $h(X)$, but also the class prediction $c(X)$ (obviously!). Thus it is more reasonable to talk about the conditional probability of $Y = c(X)$, given what is reported, that is *both* $c(X)$ and $h(X)$. We make a small but critical change to (1); we say that $(c, h)$ is *top-label calibrated* if

$$P(Y = c(X) \mid h(X), c(X)) = h(X). \tag{2}$$

(See the disambiguating Remark 2 on terminology.) Going back to the patient-disease example, top-label calibration would tell patient P the following: "among all patients, who *(just like you)* are predicted to have disease D with probability 0.6, the fraction who actually have disease D is also 0.6." Philosophically, it makes sense to condition on what is reported—both the top label and its confidence—because that is what is known to the recipient of the information; and there is no apparent justification for *not* conditioning on both.

A commonly used metric for quantifying the miscalibration of a model is the expected-calibration-error (ECE) metric. The ECE associated with confidence calibration is defined as

$$\text{conf-ECE}(c, h) := \mathbb{E}_X \left| P(Y = c(X) \mid h(X)) - h(X) \right|. \tag{3}$$

We define top-label-ECE (TL-ECE) in an analogous fashion, but also condition on $c(X)$:

$$\text{TL-ECE}(c, h) := \mathbb{E}_X \left| P(Y = c(X) \mid c(X), h(X)) - h(X) \right|. \tag{4}$$

Higher values of ECE indicate worse calibration performance. The predictor in Example 1 has conf-ECE$(c, h) = 0$. However, it has TL-ECE$(c, h) = 0.4$, revealing its miscalibration. More generally, it can be deduced as a straightforward consequence of Jensen's inequality that conf-ECE$(c, h)$ is always smaller than the TL-ECE$(c, h)$ (see Proposition 4 in Appendix H). As illustrated by Example 1, the difference can be significant. In the following subsection we illustrate that the difference can be significant on a real dataset as well. First, we make a couple of remarks.

**Remark 1** (ECE estimation using binning)**.** Estimating the ECE requires estimating probabilities conditional on some prediction such as $h(x)$. A common strategy to do this is to *bin* together nearby values of $h(x)$ using *binning* schemes (Nixon et al., 2020, Section 2.1), and compute a single estimate for the predicted and true probabilities using all the points in a bin. The calibration method we espouse in this work, histogram binning (HB), produces discrete predictions whose ECE can be estimated without further binning. Based on this, we use the following experimental protocol: we report unbinned ECE estimates while assessing HB, and binned ECE estimates for all other compared methods, which are continuous output methods (deep-nets, temperature scaling, etc). It is commonly understood that binning leads to underestimation of the effective ECE (Vaicenavicius et al., 2019; Kumar et al., 2019). Thus, using unbinned ECE estimates for HB gives HB a disadvantage compared to the binned ECE estimates we use for other methods. (This further strengthens our positive results for HB.) The binning scheme we use is equal-width binning, where the interval $[0, 1]$ is divided into $B$ equal-width intervals. Equal-width binning typically leads to lower ECE estimates compared to adaptive-width binning (Nixon et al., 2020).

**Remark 2** (Terminology)**.** The term conf-ECE was introduced by Kull et al. (2019). Most works refer to conf-ECE as just ECE (Guo et al., 2017; Nixon et al., 2020; Mukhoti et al., 2020; Kumar et al., 2018). However, some papers refer to conf-ECE as top-label-ECE (Kumar et al., 2019; Zhang et al., 2020), resulting in two different terms for the same concept. We call the older notion as conf-ECE, and *our definition of top-label calibration/ECE* (4) *is different from previous ones.*

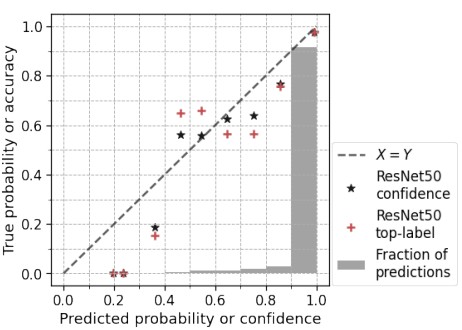

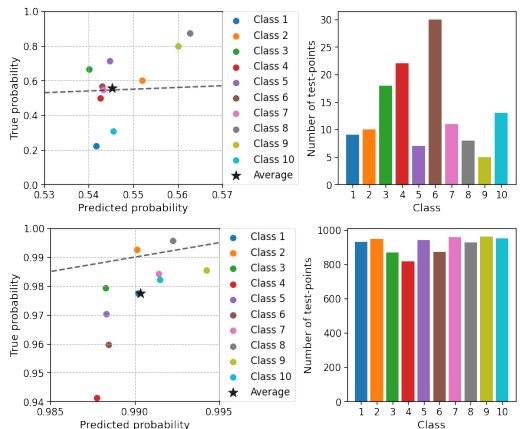

(a) Confidence reliability diagram (points marked ★) and top-label reliability diagram (points marked +) for a ResNet-50 model on the CIFAR-10 dataset; see further details in points (a) and (b) below. The **gray bars** denote the fraction of predictions in each bin. The confidence reliability diagram (mistakenly) suggests better calibration than the top-label reliability diagram.

(b) Class-wise and zoomed-in version of Figure 1a for bin 6 (top) and bin 10 (bottom); see further details in point (c) below. The ★ markers are in the same position as Figure 1a, and denote the average predicted and true probabilities. The colored points denote the predicted and true probabilities when seen class-wise. The histograms on the right show the number of test points per class within bins 6 and 10.

Figure 1: Confidence reliability diagrams misrepresent the effective miscalibration.

## 2.1 AN ILLUSTRATIVE EXPERIMENT WITH RESNET-50 ON CIFAR-10

We now compare confidence and top-label calibration using ECE estimates and reliability diagrams (Niculescu-Mizil and Caruana, 2005). This experiment can be seen as a less malignant version of Example 1. Here, confidence calibration is not completely meaningless, but can nevertheless be misleading. Figure 1 illustrates the (test-time) calibration performance of a ResNet-50 model (He et al., 2016) on the CIFAR-10 dataset (Krizhevsky, 2009). In the following summarizing points, the $(c, h)$ correspond to the ResNet-50 model.

(a) The ★ markers in Figure 1a form the **confidence reliability diagram** (Guo et al., 2017), constructed as follows. First, the $h(x)$ values on the test set are binned into one of $B = 10$ bins, $[0, 0.1), [0.1, 0.2), \dots, [0.9, 1]$, depending on the interval to which $h(x)$ belongs. The gray bars in Figure 1a indicate the fraction of $h(x)$ values in each bin—nearly 92% points belong to bin $[0.9, 1]$ and no points belong to bin $[0, 0.1)$. Next, for every bin $b$, we plot ★ $= (\text{conf}_b, \text{acc}_b)$, which are the plugin estimates of $\mathbb{E}\left[h(X) \mid h(X) \in \text{Bin } b\right]$ and $P(Y = c(X) \mid h(X) \in \text{Bin } b)$ respectively. The dashed $X = Y$ line indicates perfect confidence calibration.

(b) The + markers in Figure 1a form the **top-label reliability diagram**. Unlike the confidence reliability diagram, the top-label reliability diagram shows the average *miscalibration* across classes in a given bin. For a given class $l$ and bin $b$, define

$$\Delta_{b,l} := |\widehat{P}(Y = c(X) \mid c(X) = l, h(X) \in \text{Bin } b) - \widehat{\mathbb{E}}\left[h(X) \mid c(X) = l, h(X) \in \text{Bin } b\right]|,$$

where $\widehat{P}, \widehat{\mathbb{E}}$ denote empirical estimates based on the test data. The overall miscalibration is then

$$\Delta_b := \text{Weighted-average}(\Delta_{b,l}) = \sum_{l \in [L]} \widehat{P}(c(X) = l \mid h(X) \in \text{Bin } b) \, \Delta_{b,l}.$$

Note that $\Delta_b$ is always non-negative and does not indicate whether the overall miscalibration occurs due to under- or over-confidence; also, if the absolute-values were dropped from $\Delta_{b,l}$, then $\Delta_b$ would simply equal $\text{acc}_b - \text{conf}_b$. In order to plot $\Delta_b$ in a reliability diagram, we obtain the direction for the corresponding point from the confidence reliability diagram. Thus for every ★ $= (\text{conf}_b, \text{acc}_b)$, we plot + $= (\text{conf}_b, \text{conf}_b + \Delta_b)$ if $\text{acc}_b > \text{conf}_b$ and + $= (\text{conf}_b, \text{conf}_b - \Delta_b)$ otherwise, for every $b$. This scatter plot of the +'s gives us the top-label reliability diagram.

Figure 1a shows that there is a **visible increase in miscalibration when going from confidence calibration to top-label calibration**. To understand why this change occurs, Figure 1b zooms into the sixth bin ($h(X) \in [0.5, 0.6)$) and bin 10 ($h(X) \in [0.9, 1.0]$), as described next.

(c) Figure 1b displays the **class-wise top-label reliability diagrams** for bins 6 and 10. Note that for bin 6, the ★ marker is nearly on the $X = Y$ line, indicating that the overall accuracy matches the

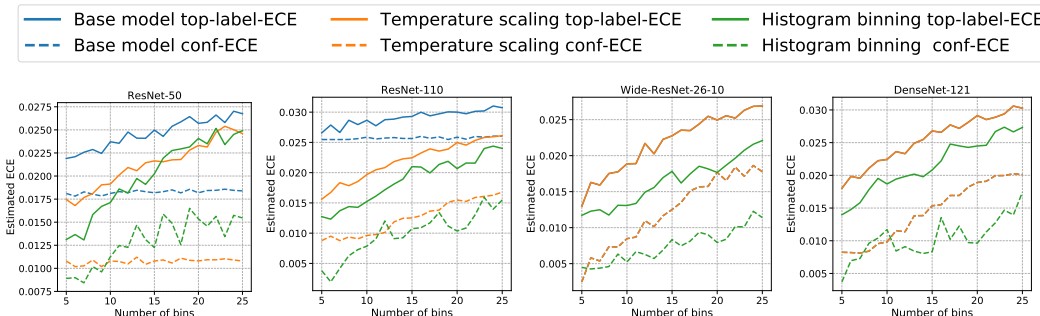

Figure 2: Conf-ECE (dashed lines) and TL-ECE (solid lines) of four deep-net architectures on CIFAR-10, as well as with recalibration using histogram binning and temperature scaling. The TL-ECE is often 2-3 times the conf-ECE, depending on the number of bins used to estimate ECE, and the architecture. Top-label histogram binning typically performs better than temperature scaling.

overall confidence of $0.545$. However, the true accuracy when class 1 was predicted is $\approx 0.2$ and the true accuracy when class 8 was predicted is $\approx 0.9$ (a very similar scenario to Example 1). For bin 10, the ★ marker indicates a miscalibration of $\approx 0.01$; however, when class 4 was predicted (roughly 8% of all test-points) the miscalibration is $\approx 0.05$.

Figure 2 displays the aggregate effect of the above phenomenon (across bins and classes) through estimates of the conf-ECE and TL-ECE. The precise experimental setup is described in Section 4. These plots display the ECE estimates of the base model, as well as the base model when recalibrated using temperature scaling (Guo et al., 2017) and our upcoming formulation of top-label histogram binning (Section 3). Since ECE estimates depend on the number of bins $B$ used (see Roelofs et al. (2020) for empirical work around this), we plot the ECE estimate for every value $B \in [5, 25]$ in order to obtain clear and unambiguous results. We find that the TL-ECE is significantly higher than the conf-ECE for most values of $B$, the architectures, and the pre- and post- recalibration models. This figure also previews the performance of our forthcoming top-label histogram binning algorithm. Top-label HB has smaller estimated TL-ECE than temperature scaling for most values of $B$ and the architectures. Except for ResNet-50, the conf-ECE estimates are also better.

To summarize, top-label calibration captures the intuition of confidence calibration by focusing on the predicted class. However, top-label calibration also conditions on the predicted class, which is always part of the prediction in any practical setting. Further, TL-ECE estimates can be substantially different from conf-ECE estimates. Thus, while it is common to compare predictors based on the conf-ECE, the TL-ECE comparison is more meaningful, and can potentially be different.

## 3 CALIBRATION ALGORITHMS FROM CALIBRATION METRICS

In this section, we unify a number of notions of multiclass calibration as multiclass-to-binary (or M2B) notions, and propose a general-purpose calibration algorithm that achieves the corresponding M2B notion of calibration. The M2B framework yields multiple novel post-hoc calibration algorithms, each of which is tuned to a specific M2B notion of calibration.

### 3.1 MULTICLASS-TO-BINARY (M2B) NOTIONS OF CALIBRATION

In Section 2, we defined confidence calibration (1) and top-label calibration (2). These notions verify calibration claims for the highest predicted probability. Other popular notions of calibration verify calibration claims for other entries in the full $L$-dimensional prediction vector. A predictor $\mathbf{h} = (h_1, h_2, \ldots, h_L)$ is said to be class-wise calibrated (Kull et al., 2017) if

$$(\text{class-wise calibration}) \quad \forall l \in [L], \, P(Y = l \mid h_l(X)) = h_l(X). \tag{5}$$

Another recently proposed notion is top-$K$ confidence calibration (Gupta et al., 2021). For some $l \in [L]$, let $c^{(l)} : \mathcal{X} \to [L]$ denote the $l$-th highest class prediction, and let $h^{(l)} : \mathcal{X} \to [L]$ denote the confidence associated with it ($c = c^{(1)}$ and $h = h^{(1)}$ are special cases). For a given $K \leqslant L$,

$$(\text{top-}K\text{-confidence calibration}) \quad \forall k \in [K], \, P(Y = c^{(k)}(X) \mid h^{(k)}(X)) = h^{(k)}(X). \tag{6}$$

| Calibration notion | Quantifier | Prediction ($\mathrm{pred}(X)$) | Binary calibration statement |
|---|---|---|---|
| Confidence | - | $h(X)$ | $P(Y = c(X) \mid \mathrm{pred}(X)) = h(X)$ |
| Top-label | - | $c(X), h(X)$ | $P(Y = c(X) \mid \mathrm{pred}(X)) = h(X)$ |
| Class-wise | $\forall l \in [L]$ | $h_l(X)$ | $P(Y = l \mid \mathrm{pred}(X)) = h_l(X)$ |
| Top-$K$-confidence | $\forall k \in [K]$ | $h^{(k)}(X)$ | $P(Y = c^{(k)}(X) \mid \mathrm{pred}(X)) = h^{(k)}(X)$ |
| Top-$K$-label | $\forall k \in [K]$ | $c^{(k)}(X), h^{(k)}(X)$ | $P(Y = c^{(k)}(X) \mid \mathrm{pred}(X)) = h^{(k)}(X)$ |

Table 1: Multiclass-to-binary (M2B) notions internally verify one or more binary calibration statements/claims. The statements in the rightmost column are required to hold almost surely.

As we did in Section 2 for confidence→top-label, top-$K$-confidence calibration can be modified to the more interpretable top-$K$-label calibration by further conditioning on the predicted labels:

$$\text{(top-}K\text{-label calibration)} \quad \forall k \in [K], P(Y = c^{(k)}(X) \mid h^{(k)}(X), c^{(k)}(X)) = h^{(k)}(X). \quad (7)$$

Each of these notions reduce multiclass calibration to one or more binary calibration requirements, where each binary calibration requirement corresponds to **verifying if the distribution of $Y$, conditioned on some prediction $\mathrm{pred}(X)$, satisfies a single binary calibration claim associated with $\mathrm{pred}(X)$**. Table 1 illustrates how the calibration notions discussed so far internally verify a number of binary calibration claims, making them M2B notions. For example, for class-wise calibration, for every $l \in [L]$, the conditioning is on $\mathrm{pred}(X) = h_l(X)$, and a single binary calibration statement is verified: $P(Y = l \mid \mathrm{pred}(X)) = h_l(X)$. Based on this property, we call each of these notions multiclass-to-binary or M2B notions.

The notion of canonical calibration mentioned in the introduction is *not* an M2B notion. Canonical calibration is discussed in detail in Appendix G. Due to the conditioning on a multi-dimensional prediction, non-M2B notions of calibration are harder to achieve or verify. For the same reason, it is possibly easier for humans to interpret binary calibration claims when taking decisions/actions.

## 3.2 ACHIEVING M2B NOTIONS OF CALIBRATION USING M2B CALIBRATORS

The M2B framework illustrates how multiclass calibration can typically be viewed via a reduction to binary calibration. The immediate consequence of this reduction is that one can now solve multiclass calibration problems by leveraging the well-developed methodology for binary calibration.

The upcoming M2B calibrators belong to the standard recalibration or post-hoc calibration setting. In this setting, one starts with a fixed pre-learnt base model $\mathbf{g} : \mathcal{X} \to \Delta^{L-1}$. The base model $\mathbf{g}$ can correspond to a deep-net, a random forest, or any 1-v-all (one-versus-all) binary classification model such as logistic regression. The base model is typically optimized for classification accuracy and may not be calibrated. The goal of post-hoc calibration is to use some given *calibration data* $\mathcal{D} = (X_1, Y_1), (X_2, Y_2), \ldots, (X_n, Y_n) \in (\mathcal{X} \times [L])^n$, typically data on which $\mathbf{g}$ was not learnt, to recalibrate $\mathbf{g}$. In practice, the calibration data is usually the same as the validation data.

To motivate M2B calibrators, suppose we want to verify if $\mathbf{g}$ is calibrated on a certain test set, based on a given M2B notion of calibration. Then, the verifying process will split the test data into a number of sub-datasets, each of which will verify one of the binary calibration claims. In Appendix A.2, we argue that the calibration data can also be viewed as a test set, and every step in the verification process can be used to provide a signal for improving calibration.

M2B calibrators take the form of *wrapper* methods that work on top of a given binary calibrator. Denote an arbitrary black-box binary calibrator as $\mathcal{A}_{\{0,1\}} : [0,1]^{\mathcal{X}} \times (\mathcal{X} \times \{0,1\})^\star \to [0,1]^{\mathcal{X}}$, where the first argument is a mapping $\mathcal{X} \to [0,1]$ that denotes a (miscalibrated) binary predicor, and the second argument is a calibration data sequence of arbitrary length. The output is a (better calibrated) binary predictor. Examples of $\mathcal{A}_{\{0,1\}}$ are histogram binning (Zadrozny and Elkan, 2001), isotonic regression (Zadrozny and Elkan, 2002), and Platt scaling (Platt, 1999). In the upcoming descriptions, we use the indicator function $\mathbb{1}\{a = b\} \in \{0,1\}$ which takes the value 1 if $a = b$, and 0 if $a \neq b$.

The general formulation of our M2B calibrator is delayed to Appendix A since the description is a bit involved. To ease readability and adhere to the space restrictions, in the main paper we describe the calibrators corresponding to top-label, class-wise, and confidence calibration (Algorithms 1–3). Each of these calibrators are different from the *classical* M2B calibrator (Algorithm 4) that has been used by Zadrozny and Elkan (2002), Guo et al. (2017), Kull et al. (2019), and most other papers

M2B calibrators: Post-hoc multiclass calibration using binary calibrators

**Input in each case:** Binary calibrator $\mathcal{A}_{\{0,1\}} : [0,1]^{\mathcal{X}} \times (\mathcal{X} \times \{0,1\})^{\star} \to [0,1]^{\mathcal{X}}$, base multiclass predictor $\mathbf{g} : \mathcal{X} \to \Delta^{L-1}$, calibration data $\mathcal{D} = (X_1, Y_1), \ldots, (X_n, Y_n)$.

---

**Algorithm 1:** Confidence calibrator

1 $c \leftarrow$ classifier or top-class based on $\mathbf{g}$;
2 $g \leftarrow$ top-class-probability based on $\mathbf{g}$;
3 $\mathcal{D}' \leftarrow \{(X_i, \mathbb{1}\{Y_i = c(X_i)\}) : i \in [n]\}$;
4 $h \leftarrow \mathcal{A}_{\{0,1\}}(g, \mathcal{D}')$;
5 **return** $(c, h)$;

---

**Algorithm 3:** Class-wise calibrator

1 Write $\mathbf{g} = (g_1, g_2, \ldots, g_L)$;
2 **for** $l \leftarrow 1$ **to** $L$ **do**
3 $\quad$ $\mathcal{D}_l \leftarrow \{(X_i, \mathbb{1}\{Y_i = l\}) : i \in [n]\}$;
4 $\quad$ $h_l \leftarrow \mathcal{A}_{\{0,1\}}(g_l, \mathcal{D}_l)$;
5 **end**
6 **return** $(h_1, h_2, \ldots, h_L)$;

---

**Algorithm 2:** Top-label calibrator

1 $c \leftarrow$ classifier or top-class based on $\mathbf{g}$;
2 $g \leftarrow$ top-class-probability based on $\mathbf{g}$;
3 **for** $l \leftarrow 1$ **to** $L$ **do**
4 $\quad$ $\mathcal{D}_l \leftarrow \{(X_i, \mathbb{1}\{Y_i = l\}) : c(X_i) = l\}$;
5 $\quad$ $h_l \leftarrow \mathcal{A}_{\{0,1\}}(g, \mathcal{D}_l)$;
6 **end**
7 $h(\cdot) \leftarrow h_{c(\cdot)}(\cdot)$ (predict $h_l(x)$ if $c(x) = l$);
8 **return** $(c, h)$;

---

**Algorithm 4:** Normalized calibrator

1 Write $\mathbf{g} = (g_1, g_2, \ldots, g_L)$;
2 **for** $l \leftarrow 1$ **to** $L$ **do**
3 $\quad$ $\mathcal{D}_l \leftarrow \{(X_i, \mathbb{1}\{Y_i = l\}) : i \in [n]\}$;
4 $\quad$ $\widetilde{h}_l \leftarrow \mathcal{A}_{\{0,1\}}(g_l, \mathcal{D}_l)$;
5 **end**
6 Normalize: for every $l \in [L]$,
$\quad$ $h_l(\cdot) := \widetilde{h}_l(\cdot) / \sum_{k=1}^{L} \widetilde{h}_k(\cdot)$;
7 **return** $(h_1, h_2, \ldots, h_L)$;

---

we are aware of, with the most similar one being Algorithm 3. Top-$K$-label and top-$K$-confidence calibrators are also explicitly described in Appendix A (Algorithms 6 and 7).

Top-label calibration requires that for every class $l \in [L]$, $P(Y = l \mid c(X) = l, h(X)) = h(X)$. Thus, to achieve top-label calibration, we must solve $L$ calibration problems. Algorithm 2 constructs $L$ datasets $\{\mathcal{D}_l : l \in [L]\}$ (line 4). The features in $\mathcal{D}_l$ are the $X_i$'s for which $c(X_i) = l$, and the labels are $\mathbb{1}\{Y_i = l\}$. Now for every $l \in [L]$, we calibrate $g$ to $h_l : \mathcal{X} \to [0,1]$ using $\mathcal{D}_l$ and any binary calibrator. The final probabilistic predictor is $h(\cdot) = h_{c(\cdot)}(\cdot)$ (that is, it predicts $h_l(x)$ if $c(x) = l$). The top-label predictor $c$ does not change in this process. Thus the accuracy of $(c, h)$ is the same as the accuracy of $\mathbf{g}$ irrespective of which $\mathcal{A}_{\{0,1\}}$ is used. Unlike the top-label calibrator, the confidence calibrator merges all classes together into a single dataset $\mathcal{D}' = \bigcup_{l \in [L]} \mathcal{D}_l$.

To achieve class-wise calibration, Algorithm 3 also solves $L$ calibration problems, but these correspond to satisfying $P(Y = l \mid h_l(X)) = h_l(X)$. Unlike top-label calibration, the dataset $D_l$ for class-wise calibration contains all the $X_i$'s (even if $c(X_i) \neq l$), and $h_l$ is passed to $\mathcal{A}_{\{0,1\}}$ instead of $h$. Also, unlike confidence calibration, $Y_i$ is replaced with $\mathbb{1}\{Y_i = l\}$ instead of $\mathbb{1}\{Y_i = c(X_i)\}$. The overall process is similar to reducing multiclass classification to $L$ 1-v-all binary classification problem, but our motivation is intricately tied to the notion of class-wise calibration.

Most popular empirical works that have discussed binary calibrators for multiclass calibration have done so using the normalized calibrator, Algorithm 4. This is almost identical to Algorithm 3, except that there is an additional normalization step (line 6 of Algorithm 4). This normalization was first proposed by Zadrozny and Elkan (2002, Section 5.2), and has been used unaltered by most other works[1] where the goal has been to simply compare direct multiclass calibrators such as temperature scaling, Dirichlet scaling, etc., to a calibrator based on binary methods (for instance, see Section 4.2 of Guo et al. (2017)). In contrast to these papers, we investigate multiple M2B reductions in an effort to identify the right reduction of multiclass calibration to binary calibration.

To summarize, the M2B characterization immediately yields a novel and different calibrator for every M2B notion. In the following section, we instantiate M2B calibrators on the binary calibrator of histogram binning (HB), leading to two new algorithms: top-label-HB and class-wise-HB, that achieve strong empirical results and satisfy distribution-free calibration guarantees.

---

[1] the only exception we are aware of is the recent work of Patel et al. (2021) who also suggest skipping normalization (see their Appendix A1); however they use a common I-Max binning scheme across classes, whereas in Algorithm 3 the predictor $h_l$ for each class is learnt completely independently of other classes

| Metric | Dataset | Architecture | Base | TS | VS | DS | N-HB | TL-HB |
|--------|---------|--------------|------|------|------|------|------|-------|
| Top-label-ECE | CIFAR-10 | ResNet-50 | 0.025 | 0.022 | 0.020 | 0.019 | **0.018** | 0.020 |
| | | ResNet-110 | 0.029 | 0.022 | 0.021 | 0.021 | **0.020** | 0.021 |
| | | WRN-26-10 | 0.023 | 0.023 | 0.019 | 0.021 | **0.012** | 0.018 |
| | | DenseNet-121 | 0.027 | 0.027 | 0.020 | 0.020 | **0.019** | 0.021 |
| | CIFAR-100 | ResNet-50 | 0.118 | 0.114 | 0.113 | 0.322 | **0.081** | 0.143 |
| | | ResNet-110 | 0.127 | 0.121 | 0.115 | 0.353 | **0.093** | 0.145 |
| | | WRN-26-10 | 0.103 | 0.103 | 0.100 | 0.304 | **0.070** | 0.129 |
| | | DenseNet-121 | 0.110 | 0.110 | 0.109 | 0.322 | **0.086** | 0.139 |
| Top-label-MCE | CIFAR-10 | ResNet-50 | 0.315 | 0.305 | 0.773 | 0.282 | 0.411 | **0.107** |
| | | ResNet-110 | 0.275 | 0.227 | 0.264 | 0.392 | 0.195 | **0.077** |
| | | WRN-26-10 | 0.771 | 0.771 | 0.498 | 0.325 | 0.140 | **0.071** |
| | | DenseNet-121 | 0.289 | 0.289 | 0.734 | 0.294 | 0.345 | **0.087** |
| | CIFAR-100 | ResNet-50 | 0.436 | 0.300 | **0.251** | 0.619 | 0.397 | 0.291 |
| | | ResNet-110 | 0.313 | **0.255** | 0.277 | 0.557 | 0.266 | 0.257 |
| | | WRN-26-10 | 0.273 | **0.255** | 0.256 | 0.625 | 0.287 | 0.280 |
| | | DenseNet-121 | 0.279 | **0.231** | 0.235 | 0.600 | 0.320 | 0.289 |

Table 2: Top-label-ECE and top-label-MCE for deep-net models (above: 'Base') and various post-hoc calibrators: temperature-scaling (TS), vector-scaling (VS), Dirichlet-scaling (DS), top-label-HB (TL-HB), and normalized-HB (N-HB). Best performing method in each row is in **bold**.

| Metric | Dataset | Architecture | Base | TS | VS | DS | N-HB | CW-HB |
|--------|---------|--------------|------|------|------|------|------|-------|
| Class-wise-ECE $\times 10^2$ | CIFAR-10 | ResNet-50 | 0.46 | 0.42 | 0.35 | 0.35 | 0.50 | **0.28** |
| | | ResNet-110 | 0.59 | 0.50 | 0.42 | 0.38 | 0.53 | **0.27** |
| | | WRN-26-10 | 0.44 | 0.44 | 0.35 | 0.39 | 0.39 | **0.28** |
| | | DenseNet-121 | 0.46 | 0.46 | **0.36** | **0.36** | 0.48 | **0.36** |
| | CIFAR-100 | ResNet-50 | 0.22 | 0.20 | 0.20 | 0.66 | 0.23 | **0.16** |
| | | ResNet-110 | 0.24 | 0.23 | 0.21 | 0.72 | 0.24 | **0.16** |
| | | WRN-26-10 | 0.19 | 0.19 | 0.18 | 0.61 | 0.20 | **0.14** |
| | | DenseNet-121 | 0.20 | 0.21 | 0.19 | 0.66 | 0.24 | **0.16** |

Table 3: Class-wise-ECE for deep-net models and various post-hoc calibrators. All methods are same as Table 2, except TL-HB is replaced with class-wise-HB (CW-HB).

## 4 EXPERIMENTS: M2B CALIBRATION WITH HISTOGRAM BINNING

Histogram binning or HB was proposed by Zadrozny and Elkan (2001) with strong empirical results for binary calibration. In HB, a base binary calibration model $g : \mathcal{X} \to [0, 1]$ is used to partition the calibration data into a number of bins so that each bin has roughly the same number of points. Then, for each bin, the probability of $Y = 1$ is estimated using the empirical distribution on the calibration data. This estimate forms the new calibrated prediction for that bin. Recently, Gupta and Ramdas (2021) showed that HB satisfies strong distribution-free calibration guarantees, which are otherwise impossible for scaling methods (Gupta et al., 2020).

Despite these results for binary calibration, studies for multiclass calibration have reported that HB typically performs worse than scaling methods such as temperature scaling (TS), vector scaling (VS), and Dirichlet scaling (DS) (Kull et al., 2019; Roelofs et al., 2020; Guo et al., 2017). In our experiments, we find that the issue is not HB but the M2B wrapper used to produce the HB baseline. With the right M2B wrapper, HB beats TS, VS, and DS. A number of calibrators have been proposed recently (Zhang et al., 2020; Rahimi et al., 2020; Patel et al., 2021; Gupta et al., 2021), but VS and DS continue to remain strong baselines which are often close to the best in these papers. We do not compare to each of these calibrators; our focus is on the M2B reduction and the message that the baselines dramatically improve with the right M2B wrapper.

We use three metrics for comparison: the first is top-label-ECE or TL-ECE (defined in (4)), which we argued leads to a more meaningful comparison compared to conf-ECE. Second, we consider the more stringent maximum-calibration-error (MCE) metric that assesses the worst calibration across predictions (see more details in Appendix E.3). For top-label calibration MCE is given by $\text{TL-MCE}(c, h) := \max_{l \in [L]} \sup_{r \in \text{Range}(h)} |P(Y = l \mid c(X) = l, h(X) = r) - r|$. To assess class-wise calibration, we use class-wise-ECE defined as the average calibration error across classes:

CW-ECE$(c, \mathbf{h}) := L^{-1} \sum_{l=1}^{L} \mathbb{E}_X |P(Y = l \mid h_l(X)) - h_l(X)|$. All ECE/MCE estimation is performed as described in Remark 1. For further details, see Appendix E.2.

**Formal algorithm and theoretical guarantees.** Top-label-HB (TL-HB) and class-wise-HB (CW-HB) are explicitly stated in Appendices B and C respectively; these are instantiations of the top-label calibrator and class-wise calibrator with HB. N-HB is the the normalized calibrator (Algorithm 4) with HB, which is the same as CW-HB, but with an added normalization step. In the Appendix, we extend the binary calibration guarantees of Gupta and Ramdas (2021) to TL-HB and CW-HB (Theorems 1 and 2). We informally summarize one of the results here: if there are at least $k$ calibration points-per-bin, then the expected-ECE is bounded as: $\mathbb{E}\left[\text{(TL-) or (CW-) ECE}\right] \leqslant \sqrt{1/2k}$, for TL-HB and CW-HB respectively. The outer $\mathbb{E}$ above is an expectation over the calibration data, and corresponds to the randomness in the predictor learnt on the calibration data. Note that the ECE itself is an expected error over an unseen i.i.d. test-point $(X, Y) \sim P$.

**Experimental details.** We experimented on the CIFAR-10 and CIFAR-100 datasets, which have 10 and 100 classes each. The base models are deep-nets with the following architectures: ResNet-50, Resnet-110, Wide-ResNet-26-10 (WRN) (Zagoruyko and Komodakis, 2016), and DenseNet-121 (Huang et al., 2017). Both CIFAR datasets consist of 60K (60,000) points, which are split as 45K/5K/10K to form the train/validation/test sets. The validation set was used for post-hoc calibration and the test set was used for evaluation through ECE/MCE estimates. Instead of training new models, we used the pre-trained models of Mukhoti et al. (2020). We then ask: *"which post-hoc calibrator improves the calibration the most?"* We used their Brier score and focal loss models in our experiments (Mukhoti et al. (2020) report that these are the empirically best performing loss functions). *All results in the main paper are with Brier score, and results with focal loss are in Appendix E.4.* Implementation details for TS, VS, and DS are in Appendix E.

**Findings.** In Table 2, we report the binned ECE and MCE estimates when $B = 15$ bins are used by HB, and for ECE estimation. We make the following observations:

(a) For TL-ECE, N-HB is the best performing method for both CIFAR-10 and CIFAR-100. While most methods perform similarly across architectures for CIFAR-10, there is high variation in CIFAR-100. DS is the worst performing method on CIFAR-100, but TL-HB also performs poorly. We believe that this could be because the data splitting scheme of the TL-calibrator (line 4 of Algorithm 2) splits datasets across the predicted classes, and some classes in CIFAR-100 occur very rarely. This is further discussed in Appendix E.6.

(b) For TL-MCE, TL-HB is the best performing method on CIFAR-10, by a huge margin. For CIFAR-100, TS or VS perform slightly better than TL-HB. Since HB ensures that each bin gets roughly the same number of points, the predictions are well calibrated across bins, leading to smaller TL-MCE. A similar observation was also made by Gupta and Ramdas (2021).

(c) For CW-ECE, CW-HB is the best performing method across the two datasets and all four architectures. The N-HB method which has been used in many CW-ECE baseline experiments performs terribly. In other words, skipping the normalization step leads to a large improvement in CW-ECE. **This observation is one of our most striking findings.** To shed further light on this, we note that the distribution-free calibration guarantees for CW-HB shown in Appendix C no longer hold post-normalization. Thus, both our theory and experiments indicate that skipping normalization improves CW-ECE performance.

**Additional experiments in the Appendix.** In Appendix E.5, we report each of the results in Tables 2 and 3 with the number of bins taking every value in the range $[5, 25]$. Most observations remain the same under this expanded study. In Appendix B.2, we consider top-label calibration for the class imbalanced COVTYPE-7 dataset, and show that TL-HB adapts to tail/infrequent classes.

## 5 CONCLUSION

We make two contributions to the study of multiclass calibration: (i) defining the new notion of top-label calibration which enforces a natural minimal requirement on a multiclass predictor—the probability score for the top class prediction should be calibrated; (ii) developing a multiclass-to-binary (M2B) framework which posits that various notions of multiclass calibration can be achieved via reduction to binary calibration, balancing practical utility with statistically tractability. Since it is important to identify appropriate notions of calibration in any structured output space (Kuleshov et al., 2018; Gneiting et al., 2007), we anticipate that the philosophy behind the M2B framework could find applications in other structured spaces.

## 6    REPRODUCIBILITY STATEMENT

Some reproducibility desiderata, such as external code and libraries that were used are summarized in Appendix E.1. All code to generate results with the CIFAR datasets is attached in the supplementary material. Our base models were pre-trained deep-net models generated by Mukhoti et al. (2020), obtained from `www.robots.ox.ac.uk/∼viveka/focal_calibration/` (corresponding to 'brier_score' and 'focal_loss_adaptive_53' at the above link). By avoiding training of new deep-net models with multiple hyperparameters, we also consequently avoided selection biases that inevitably creep in due to test-data-peeking. The predictions of the pre-trained models were obtained using the code at `https://github.com/torrvision/focal_calibration`.

## 7    ETHICS STATEMENT

Post-hoc calibration is a post-processing step that can be applied on top of miscalibrated machine learning models to increase their reliability. As such, we believe our work should improve the transparency and explainability of machine learning models. However, we outline a few limitations. Post-hoc calibration requires keeping aside a fresh, representative dataset, that was not used for training. If this dataset is too small, the resulting calibration guarantee can be too weak to be meaningful in practice. Further, if the test data distribution shifts in significant ways, additional corrections may be needed to recalibrate (Gupta et al., 2020; Podkopaev and Ramdas, 2021). A well calibrated classifier is not necessarily an accurate or a fair one, and vice versa (Kleinberg et al., 2017). Deploying calibrated models in critical applications like medicine, criminal law, banking, etc. does not preclude the possibility of the model being frequently wrong or unfair.

## ACKNOWLEDGEMENTS

This work used the Extreme Science and Engineering Discovery Environment (XSEDE), which is supported by National Science Foundation grant number ACI-1548562 (Towns et al., 2014). Specifically, it used the Bridges-2 system, which is supported by NSF award number ACI-1928147, at the Pittsburgh Supercomputing Center (PSC). CG's research was supported by the generous Bloomberg Data Science Ph.D. Fellowship. CG would like to thank Saurabh Garg and Youngseog Chung for interesting discussions, and Viveka Kulharia for help with the focal calibration repository. Finally, we thank Zack Lipton, the ICLR reviewers, and the ICLR area chair, for excellent feedback that helped improve the writing of the paper.

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

---

**Algorithm 5:** Post-hoc calibrator for a given M2B calibration notion $\mathcal{C}$

---

**Input:** Base (uncalibrated) multiclass predictor **g**, calibration data
$\mathcal{D} = (X_1, Y_1), \ldots, (X_n, Y_n)$, binary calibrator
$\mathcal{A}_{\{0,1\}} : [0,1]^{\mathcal{X}} \times (\mathcal{X} \times \{0,1\})^{\star} \to [0,1]^{\mathcal{X}}$

1   $K \leftarrow$ number of distinct calibration claims that $\mathcal{C}$ verifies;
2   **for** *each claim $k \in [K]$* **do**
3      From **g**, infer $(\widetilde{c}, \widetilde{g}) \leftarrow$ (label-predictor, probability-predictor) corresponding to claim $k$;
4      $\mathcal{D}_k \leftarrow \{(X_i, Z_i)\}$, where $Z_i \leftarrow \mathbb{1}\{Y_i = \widetilde{c}(X_i)\}$;
5      **if** *conditioning does not include class prediction $\widetilde{c}$* **then**
6          — (confidence, top-$K$-confidence, and class-wise calibration) —
7          $h_k \leftarrow \mathcal{A}_{\{0,1\}}(\widetilde{g}, \mathcal{D}_k)$;
8      **end**
9      **else**
10          — (top-label and top-$K$-label calibration) —
11          **for** $l \in [L]$ **do**
12              $\mathcal{D}_{k,l} \leftarrow \{(X_i, Z_i) \in \mathcal{D}_k : \widetilde{c}(X_i) = l\}$;
13              $h_{k,l} \leftarrow \mathcal{A}_{\{0,1\}}(\widetilde{g}, \mathcal{D}_{k,l})$;
14          **end**
15          $h_k(\cdot) \leftarrow h_{k,\widetilde{c}(\cdot)}(\cdot)$   ($h_k$ predicts $h_{k,l}(x)$ if $\widetilde{c}(x) = l$);
16      **end**
17   **end**
18   — (the new predictor replaces each $\widetilde{g}$ with the corresponding $h_k$) —
19   **return** (label-predictor, $h_k$) corresponding to each claim $k \in [K]$;

---

**Input for Algorithms 6 and 7:** base multiclass predictor $\mathbf{g} : \mathcal{X} \to \Delta^{L-1}$, calibration data $\mathcal{D} = (X_1, Y_1), \ldots, (X_n, Y_n)$, binary calibrator $\mathcal{A}_{\{0,1\}} : [0,1]^{\mathcal{X}} \times (\mathcal{X} \times \{0,1\})^{\star} \to [0,1]^{\mathcal{X}}$.

---

**Algorithm 6:** Top-$K$-label calibrator

---

1   For every $k \in [K]$, infer from **g** the $k$-th largest class predictor $c^{(k)}$ and the associated probability $g^{(k)}$;
2   **for** $k \leftarrow 1$ **to** $K$ **do**
3      **for** $l \leftarrow 1$ **to** $L$ **do**
4          $\mathcal{D}_{k,l} \leftarrow \{(X_i, \mathbb{1}\{Y_i = l\}) : c^{(k)}(X_i) = l\}$;
5          $h^{(k,l)} \leftarrow \mathcal{A}_{\{0,1\}}(g^{(k)}, \mathcal{D}_{k,l})$;
6      **end**
7      $h^{(k)} \leftarrow h^{(k,c^{(k)}(\cdot))}(\cdot)$;
8   **end**
9   **return** $(h^{(1)}, h^{(2)}, \ldots, h^{(K)})$;

---

**Algorithm 7:** Top-$K$-confidence calibrator

---

1   For every $k \in [K]$, infer from **g** the $k$-th largest class predictor $c^{(k)}$ and the associated probability $g^{(k)}$;
2   **for** $k \leftarrow 1$ **to** $K$ **do**
3      $\mathcal{D}_k \leftarrow \{(X_i, \mathbb{1}\{Y_i = l\}) : i \in [n]\}$;
4      $h^{(k)} \leftarrow \mathcal{A}_{\{0,1\}}(g^{(k)}, \mathcal{D}_k)$;
5   **end**
6   **return** $(h^{(1)}, h^{(2)}, \ldots, h^{(K)})$;

---

## A   ADDENDUM TO SECTION 3 "CALIBRATION ALGORITHMS FROM CALIBRATION METRICS"

In Section 3, we introduced the concept of M2B calibration, and showed that popular calibration notions are in fact M2B notions (Table 1). We showed how the calibration notions of top-label, class-wise, and confidence calibration can be achieved using a corresponding M2B calibrator. In the following subsection, we present the general-purpose *wrapper* Algorithm 5 that can be used to derive an M2B calibrator from any given M2B calibration notion that follows the rubric specified by Table 1. In Appendix A.2, we illustrate the philosophy of M2B calibration using a simple example with a dataset that contains 6 points. This example also illustrates the top-label-calibrator, the class-wise-calibrator, and the confidence-calibrator.

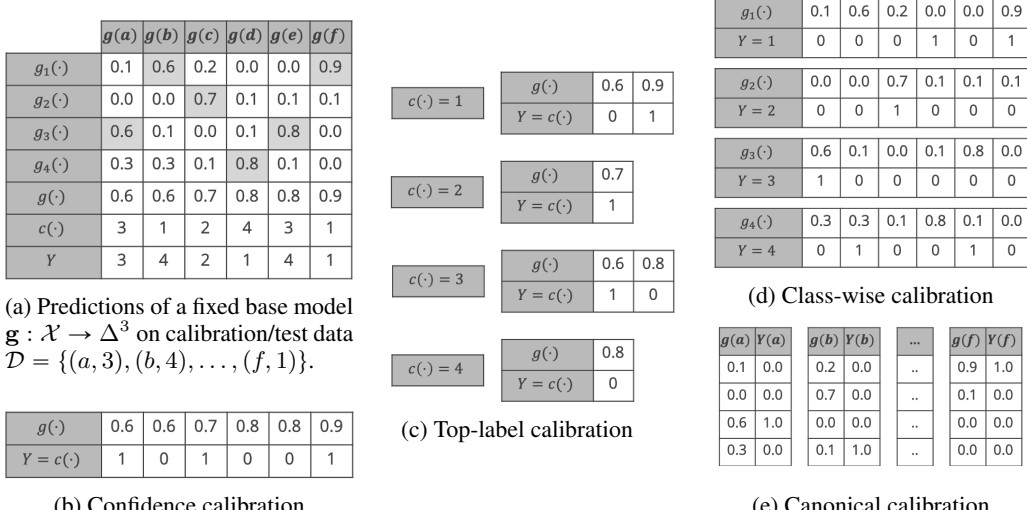

(a) Predictions of a fixed base model $\mathbf{g} : \mathcal{X} \to \Delta^3$ on calibration/test data $\mathcal{D} = \{(a, 3), (b, 4), \ldots, (f, 1)\}$.

(b) Confidence calibration

(c) Top-label calibration

(d) Class-wise calibration

(e) Canonical calibration

Figure 3: Illustrative example for Section A.2. The numbers in plot (a) correspond to the predictions made by $\mathbf{g}$ on a dataset $\mathcal{D}$. If $\mathcal{D}$ were a test set, plots (b–e) show how it should be used to verify if $\mathbf{g}$ satisfies the corresponding notion of calibration. Consequently, we argue that if $\mathcal{D}$ were a calibration set, and we want to achieve one of the notions (b–e), then the data shown in the corresponding plots should be the data used to calibrate $\mathbf{g}$ as well.

## A.1 GENERAL-PURPOSE M2B CALIBRATOR

Denote some M2B notion of calibration as $\mathcal{C}$. Suppose $\mathcal{C}$ corresponds to $K$ binary calibration claims. The outer for-loop in Algorithm 5, runs over each such claim in $\mathcal{C}$. For example, for class-wise calibration, $K = L$ and for confidence and top-label calibration, $K = 1$. Corresponding to each claim, there is a probability-predictor that the conditioning is to be done on, such as $g$ or $g_l$ or $g_{(k)}$. Additionally, there may be conditioning on the label predictor such as $c$ or $c_{(k)}$. These are denoted as $(\widetilde{c}, \widetilde{g})$ in Algorithm 5. For confidence and top-label calibration, $\widetilde{c} = c$, the top-label-confidence. For class-wise calibration, when $\widetilde{g} = g_l$, we have $\widetilde{c}(\cdot) = l$.

If there is no label conditioning in the calibration notion, such as in confidence, top-$K$-confidence, and class-wise calibration, then we enter the if-condition inside the for-loop. Here $h_k$ is learnt using a single calibration dataset and a single call to $\mathcal{A}_{\{0,1\}}$. Otherwise, if there is label conditioning, such as in top-label and top-$K$-label calibration, we enter the else-condition, where we learn a separate $h_{k,l}$ for every $l \in [L]$, using a different part of the dataset $\mathcal{D}_l$ in each case. Then $h_k(x)$ equals $h_{k,l}(x)$ if $\widetilde{c}(x) = l$.

Finally, since $\mathcal{C}$ is verifying a sequence of claims, the output of Algorithm 5 is a sequence of predictors. Each original prediction $(\widetilde{c}, \widetilde{g})$ corresponding to the $\mathcal{C}$ is replaced with $(\widetilde{c}, h_k)$. This is the output of the M2B calibrator. Note that the $\widetilde{c}$ values are not changed. This output appears abstract, but normally, it can be represented in an interpretable way. For example, for class-wise calibration, the output is just a sequence of predictors, one for each class: $(h_1, h_2, \ldots, h_L)$.

This general-purpose M2B calibrators can be used to achieve any M2B calibration notion: top-label calibration (Algorithm 2), class-wise calibration (Algorithm 3), confidence calibration (Algorithm 1), top-$K$-label calibration (Algorithm 6), and top-$K$-confidence calibration (Algorithm 7).

## A.2 AN EXAMPLE TO ILLUSTRATE THE PHILOSOPHY OF M2B CALIBRATION

Figure 3a shows the predictions of a given base model $\mathbf{g}$ on a given dataset $\mathcal{D}$. Suppose $\mathcal{D}$ is a *test set*, and we are testing confidence calibration. Then the only predictions that matter are the top-predictions corresponding to the shaded values. These are stripped out and shown in Figure 3b, in the $g(\cdot)$ row. Note that the indicator $\mathbb{1}\{Y = c(\cdot)\}$ is sufficient to test confidence calibration and given this, the $c(X)$ are not needed. Thus the second row in Figure 3b only shows these indicators.

---

**Algorithm 8:** Top-label histogram binning

---

**Input:** Base multiclass predictor $\mathbf{g}$, calibration data $\mathcal{D} = (X_1, Y_1), \ldots, (X_n, Y_n)$
**Hyperparameter:** # points per bin $k \in \mathbb{N}$ (say 50), tie-breaking parameter $\delta > 0$ (say $10^{-10}$)
**Output:** Top-label calibrated predictor $(c, h)$

1  $c \leftarrow$ classifier or top-class based on $\mathbf{g}$;
2  $g \leftarrow$ top-class-probability based on $\mathbf{g}$;
3  **for** $l \leftarrow 1$ **to** $L$ **do**
4      $\mathcal{D}_l \leftarrow \{(X_i, \mathbb{1}\{Y_i = l\}) : c(X_i) = l\}$ and $n_l \leftarrow |\mathcal{D}_l|$;
5      $h_l \leftarrow$ Binary-histogram-binning$(g, \mathcal{D}_l, \lfloor n_l/k \rfloor, \delta)$;
6  **end**
7  $h(\cdot) \leftarrow h_{c(\cdot)}(\cdot)$;
8  **return** $(c, h)$;

---

Verifying top-label calibration is similar (Figure 3c), but in addition to the predictions $g(\cdot)$, we also retain the values of $c(\cdot)$. Thus the $g(\cdot)$ and $\mathbb{1}\{Y = c(\cdot)\}$ are shown, but split across the 4 classes. Class-wise calibration requires access to all the predictions, however, each class is considered separately as indicated by Figure 3d. Canonical calibration looks at the full prediction vector in each case. However, in doing so, it becomes unlikely that $\mathbf{g}(\mathbf{x}) = \mathbf{g}(\mathbf{y})$ for any $\mathbf{x}, \mathbf{y}$ since the number of values that $\mathbf{g}$ can take is now exponential.

Let us turn this around and suppose that $\mathcal{D}$ were a calibration set instead of a test set. We argue that $\mathcal{D}$ should be used in the *same way, whether testing or calibrating*. Thus, if confidence calibration is to be achieved, we should focus on the $(g, \mathbb{1}\{Y = c(\cdot)\})$ corresponding to $\mathbf{g}$. If top-label calibration is to be achieved, we should use the $(c, g)$ values. If class-wise calibration is to be achieved, we should look at each $g_l$ separately and solve $L$ different problems. Finally, for canonical calibration, we must look at the entire $\mathbf{g}$ vector as a single unit. This is the core philosophy behind M2B calibrators: if binary claims are being verified, solve binary calibration problems.

## B  DISTRIBUTION-FREE TOP-LABEL CALIBRATION USING HISTOGRAM BINNING

In this section, we formally describe histogram binning (HB) with the top-label-calibrator (Algorithm 2) and provide methodological insights through theory and experiments.

### B.1  FORMAL ALGORITHM AND THEORETICAL GUARANTEES

Algorithm 8 describes the top-label calibrator formally using HB as the binary calibration algorithm. The function called in line 5 is Algorithm 2 of Gupta and Ramdas (2021). The first argument in the call is the top-label confidence predictor, the second argument is the dataset to be used, the third argument is the number of bins to be used, and the fourth argument is a tie-breaking parameter (described shortly). While previous empirical works on HB fixed the *number of bins per class*, the analysis of Gupta and Ramdas (2021) suggests that a more principled way of choosing the number of bins is to fix the *number of points per bin*. This is parameter $k$ of Algorithm 8. Given $k$, the number of bins is decided separately for every class as $\lfloor n_l/k \rfloor$ where $n_l$ is the number of points predicted as class $l$. This choice is particularly relevant for top-label calibration since $n_l$ can be highly non-uniform (we illustrate this empirically in Section B.2). The tie-breaking parameter $\delta$ can be arbitrarily small (like $10^{-10}$), and its significance is mostly theoretical—it is used to ensure that outputs of different bins are not exactly identical by chance, so that conditioning on a calibrated probability output is equivalent to conditioning on a bin; this leads to a cleaner theoretical guarantee.

HB recalibrates $g$ to a piecewise constant function $h$ that takes one value per bin. Consider a specific bin $b$; the $h$ value for this bin is computed as the average of the indicators $\{\mathbb{1}\{Y_i = c(X_i)\} : X_i \in \text{Bin } b\}$. This is an estimate of the *bias* of the bin $P(Y = c(X) \mid X \in \text{Bin } b)$. A concentration inequality can then be used to bound the deviation between the estimate and the true bias to prove distribution-free calibration guarantees. In the forthcoming Theorem 1, we show high-probability and in-expectation bounds on the the TL-ECE of HB. Additionally, we show marginal and condi-

tional top-label calibration bounds, defined next. These notions were proposed in the binary calibration setting by Gupta et al. (2020) and Gupta and Ramdas (2021). In the definition below, $\mathcal{A}$ refers to any algorithm that takes as input calibration data $\mathcal{D}$ and an initial classifier $\mathbf{g}$ to produce a top-label predictor $c$ and an associated probability map $h$. Algorithm 8 is an example of $\mathcal{A}$.

**Definition 1** (Marginal and conditional top-label calibration). Let $\varepsilon, \alpha \in (0, 1)$ be some given levels of approximation and failure respectively. An algorithm $\mathcal{A} : (\mathbf{g}, \mathcal{D}) \mapsto (c, h)$ is

(a) $(\varepsilon, \alpha)$-marginally top-label calibrated if for every distribution $P$ over $\mathcal{X} \times [L]$,

$$P\Big( |P(Y = c(X) \mid c(X), h(X)) - h(X)| \leqslant \varepsilon \Big) \geqslant 1 - \alpha. \qquad (8)$$

(b) $(\varepsilon, \alpha)$-conditionally top-label calibrated if for every distribution $P$ over $\mathcal{X} \times [L]$,

$$P\Big( \forall\, l \in [L], r \in \mathrm{Range}(h), |P(Y = c(X) \mid c(X) = l, h(X) = r) - r| \leqslant \varepsilon \Big) \geqslant 1 - \alpha. \qquad (9)$$

To clarify, all probabilities are taken over the test point $(X, Y) \sim P$, the calibration data $\mathcal{D} \sim P^n$, and any other inherent algorithmic randomness in $\mathcal{A}$; these are all implicit in $(c, h) = \mathcal{A}(\mathcal{D}, \mathbf{g})$. Marginal calibration asserts that with high probability, on average over the distribution of $\mathcal{D}, X$, $P(Y = c(X) \mid c(X), h(X))$ is at most $\varepsilon$ away from $h(X)$. In comparison, TL-ECE is the average of these deviations over $X$. Marginal calibration may be a more appropriate metric for calibration than TL-ECE if we are somewhat agnostic to probabilistic errors less than some fixed threshold $\varepsilon$ (like 0.05). Conditional calibration is a strictly stronger definition that requires the deviation to be at most $\varepsilon$ for every possible prediction $(l, r)$, including rare ones, not just on average over predictions. This may be relevant in medical settings where we want the prediction on every patient to be reasonably calibrated. Algorithm 8 satisfies the following calibration guarantees.

**Theorem 1.** *Fix hyperparameters $\delta > 0$ (arbitrarily small) and points per bin $k \geqslant 2$, and assume $n_l \geqslant k$ for every $l \in [L]$. Then, for any $\alpha \in (0, 1)$, Algorithm 8 is $(\varepsilon_1, \alpha)$-marginally and $(\varepsilon_2, \alpha)$-conditionally top-label calibrated for*

$$\varepsilon_1 = \sqrt{\frac{\log(2/\alpha)}{2(k-1)}} + \delta, \qquad and \qquad \varepsilon_2 = \sqrt{\frac{\log(2n/k\alpha)}{2(k-1)}} + \delta. \qquad (10)$$

*Further, for any distribution $P$ over $\mathcal{X} \times [L]$, we have $P(\textit{TL-ECE}(c, h) \leqslant \varepsilon_2) \geqslant 1 - \alpha$, and $\mathbb{E}\left[\textit{TL-ECE}(c, h)\right] \leqslant \sqrt{1/2k} + \delta$.*

The proof in Appendix H is a multiclass top-label adaption of the guarantee in the binary setting by Gupta and Ramdas (2021). The $\widetilde{O}(1/\sqrt{k})$ dependence of the bound relies on Algorithm 8 delegating at least $k$ points to every bin. Since $\delta$ can be chosen to be arbitrarily small, setting $k = 50$ gives roughly $\mathbb{E}_{\mathcal{D}}\left[\text{TL-ECE}(h)\right] \leqslant 0.1$. Base on this, we suggest setting $k \in [50, 150]$ in practice.

## B.2 Top-label histogram binning adapts to class imbalanced datasets

The principled methodology of fixing the number of points per bin reaps practical benefits. Figure 4 illustrates this through the performance of HB for the class imbalanced COVTYPE-7 dataset (Blackard and Dean, 1999) with class ratio approximately 36% for class 1 and 49% for class 2. The entire dataset has 581012 points which is divided into train-test in the ratio 70:30. Then, 10% of the training points are held out for calibration ($n = |\mathcal{D}| = 40671$). The base classifier is a random forest (RF) trained on the remaining training points (it achieves around 95% test accuracy). The RF is then recalibrated using HB. The top-label reliability diagrams in Figure 4a illustrate that the original RF (in orange) is *underconfident* on both the most likely and least likely classes. Additional figures in Appendix F show that the RF is always underconfident no matter which class is predicted as the top-label. HB (in green) recalibrates the RF effectively across all classes. Validity plots (Gupta and Ramdas, 2021) estimate how the LHS of condition (8), denoted as $V(\varepsilon)$, varies with $\varepsilon$. We observe that for all $\varepsilon$, $V(\varepsilon)$ is higher for HB. The rightmost barplot compares the estimated TL-ECE for all classes, and also shows the class proportions. While the original RF is significantly miscalibrated for

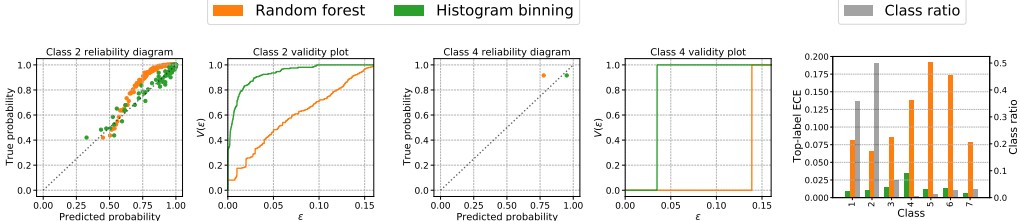

(a) Top-label histogram binning (Algorithm 8) with $k = 100$ points per bin. Class 4 has only 183 calibration points. Algorithm 8 adapts and uses only a single bin to ensure that the TL-ECE on class 4 is comparable to the TL-ECE on class 2. Overall, the random forest classifier has significantly higher TL-ECE for the least likely classes (4, 5, and 6), but the post-calibration TL-ECE using binning is quite uniform.

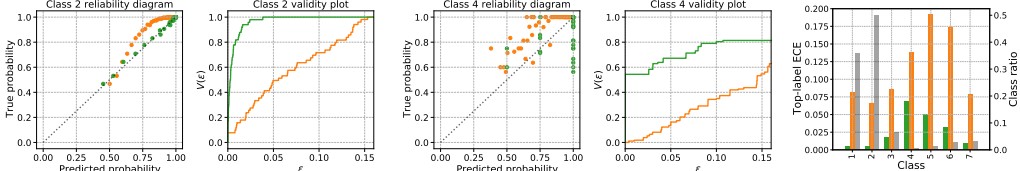

(b) Histogram binning with $B = 50$ bins for every class. Compared to Figure 4a, the post-calibration TL-ECE for the most likely classes decreases while the TL-ECE for the least likely classes increases.

Figure 4: Recalibration of a random forest using histogram binning on the class imbalanced COVTYPE-7 dataset (class 2 is roughly 100 times likelier than class 4). By ensuring a fixed number of calibration points per bin, Algorithm 8 obtains relatively uniform top-label calibration across classes (Figure 4a). In comparison, if a fixed number of bins are chosen for all classes, the performance deteriorates for the least likely classes (Figure 4b).

the less likely classes, HB has a more uniform miscalibration across classes. Figure 4b considers a slightly different HB algorithm where the number of points per class is not adapted to the number of times the class is predicted, but is fixed beforehand (this corresponds to replacing $\lfloor n_l/k \rfloor$ in line 5 of Algorithm 8 with a fixed $B \in \mathbb{N}$). While even in this setting there is a drop in the TL-ECE compared to the RF model, the final profile is less uniform compared to fixing the number of points per bin.

The validity plots and top-label reliability diagrams for all the 7 classes are reported in Figure 9 in Appendix F, along with some additional observations.

## C  DISTRIBUTION-FREE CLASS-WISE CALIBRATION USING HISTOGRAM BINNING

In this section, we formally describe histogram binning (HB) with the class-wise-calibrator (Algorithm 3) and provide theoretical guarantees for it. The overall procedure is called class-wise-HB. Further details and background on HB are contained in Appendix B, where top-label-HB is described.

### C.1  FORMAL ALGORITHM

To achieve class-wise calibration using binary routines, we learn each component function $h_l$ in a 1-v-all fashion as described in Algorithm 3. Algorithm 9 contains the pseudocode with the underlying routine as binary HB. To learn $h_l$, we use a dataset $\mathcal{D}_l$, which unlike top-label HB (Algorithm 8), contains $X_i$ even if $c(X_i) \neq l$. However the $Y_i$ is replaced with $\mathbb{1}\{Y_i = l\}$. The number of points per bin $k_l$ can be different for different classes, but generally one would set $k_1 = \ldots = k_L = k \in \mathbb{N}$. Larger values of $k_l$ will lead to smaller $\varepsilon_l$ and $\delta_l$ in the guarantees, at loss of sharpness since the number of bins $\lfloor n/k_l \rfloor$ would be smaller.

---

**Algorithm 9:** Class-wise histogram binning

---

**Input:** Base multiclass predictor $\mathbf{g} : \mathcal{X} \rightarrow \Delta^{L-1}$, calibration data $\mathcal{D} = (X_1, Y_1), \ldots, (X_n, Y_n)$
**Hyperparameter:** # points per bin $k_1, k_2, \ldots, k_l \in \mathbb{N}^L$ (say each $k_l = 50$), tie-breaking
  parameter $\delta > 0$ (say $10^{-10}$)
**Output:** $L$ class-wise calibrated predictors $h_1, h_2, \ldots, h_L$

1 **for** $l \leftarrow 1$ **to** $L$ **do**
2     $\mathcal{D}_l \leftarrow \{(X_i, \mathbb{1}\{Y_i = l\}) : i \in [n]\})\};$
3     $h_l \leftarrow$ Binary-histogram-binning$(g_l, \mathcal{D}_l, \lfloor n/k_l \rfloor, \delta);$
4 **end**
5 **return** $(h_1, h_2, \ldots, h_L);$

---

## C.2 CALIBRATION GUARANTEES

A general algorithm $\mathcal{A}$ for class-wise calibration takes as input calibration data $\mathcal{D}$ and an initial classifier $\mathbf{g}$ to produce an approximately class-wise calibrated predictor $\mathbf{h} : \mathcal{X} \rightarrow [0, 1]^L$. Define the notation $\boldsymbol{\varepsilon} = (\varepsilon_1, \varepsilon_2, \ldots, \varepsilon_L) \in (0, 1)^L$ and $\boldsymbol{\alpha} = (\alpha_1, \alpha_2, \ldots, \alpha_L) \in (0, 1)^L$.

**Definition 2** (Marginal and conditional class-wise calibration). Let $\boldsymbol{\varepsilon}, \boldsymbol{\alpha} \in (0, 1)^L$ be some given levels of approximation and failure respectively. An algorithm $\mathcal{A} : (\mathbf{g}, \mathcal{D}) \mapsto \mathbf{h}$ is

(a) $(\boldsymbol{\varepsilon}, \boldsymbol{\alpha})$-marginally class-wise calibrated if for every distribution $P$ over $\mathcal{X} \times [L]$ and for every $l \in [L]$

$$P\Big( |P(Y = l \mid h_l(X)) - h_l(X)| \leqslant \varepsilon_l \Big) \geqslant 1 - \alpha_l. \tag{11}$$

(b) $(\boldsymbol{\varepsilon}, \boldsymbol{\alpha})$-conditionally class-wise calibrated if for every distribution $P$ over $\mathcal{X} \times [L]$ and for every $l \in [L]$,

$$P\Big( \forall r \in \mathrm{Range}(h_l), |P(Y = l \mid h_l(X) = r) - r| \leqslant \varepsilon_l \Big) \geqslant 1 - \alpha_l. \tag{12}$$

Definition 2 requires that each $h_l$ is $(\varepsilon_l, \alpha_l)$ calibrated in the binary senses defined by Gupta et al. (2021, Definitions 1 and 2). From Definition 2, we can also *uniform* bounds that hold simultaneously over every $l \in [L]$. Let $\alpha = \sum_{l=1}^{L} \alpha_l$ and $\varepsilon = \max_{l \in [L]} \varepsilon_l$. Then (11) implies

$$P\Big( \forall l \in [L], |P(Y = l \mid h_l(X)) - h_l(X)| \leqslant \varepsilon \Big) \geqslant 1 - \alpha, \tag{13}$$

and (12) implies

$$P\Big( \forall l \in [L], r \in \mathrm{Range}(h_l), |P(Y = l \mid h_l(X) = r) - r| \leqslant \varepsilon \Big) \geqslant 1 - \alpha. \tag{14}$$

The choice of not including the uniformity over $L$ in Definition 2 reveals the nature of our class-wise HB algorithm and the upcoming theoretical guarantees: (a) we learn the $h_l$'s separately for each $l$ and do not combine the learnt functions in any way (such as normalization), (b) we do not combine the calibration inequalities for different $[L]$ in any other way other than a union bound. Thus the only way we can show (13) (or (14)) is by using a union bound over (11) (or (12)).

We now state the distribution-free calibration guarantees satisfied by Algorithm 9.

**Theorem 2.** *Fix hyperparameters $\delta > 0$ (arbitrarily small) and points per bin $k_1, k_2, \ldots, k_l \geqslant 2$, and assume $n_l \geqslant k_l$ for every $l \in [L]$. Then, for every $l \in [L]$, for any $\alpha_l \in (0, 1)$, Algorithm 9 is $(\boldsymbol{\varepsilon}^{(\mathbf{1})}, \boldsymbol{\alpha})$-marginally and $(\boldsymbol{\varepsilon}^{(\mathbf{2})}, \boldsymbol{\alpha})$-conditionally class-wise calibrated with*

$$\varepsilon_l^{(1)} = \sqrt{\frac{\log(2/\alpha_l)}{2(k_l - 1)}} + \delta, \qquad and \qquad \varepsilon_l^{(2)} = \sqrt{\frac{\log(2n/k_l\alpha_l)}{2(k_l - 1)}} + \delta. \tag{15}$$

*Further, for any distribution $P$ over $\mathcal{X} \times [L]$,*

   *(a) $P(\text{CW-ECE}(c, h) \leqslant \max_{l \in [L]} \varepsilon_l^{(2)}) \geqslant 1 - \sum_{l \in [L]} \alpha_l$, and*

   *(b) $\mathbb{E}[\text{CW-ECE}(c, h)] \leqslant \max_{l \in [L]} \sqrt{1/2k_l} + \delta$.*

Theorem 2 is proved in Appendix H. The proof follows by using the result of Gupta and Ramdas (2021, Theorem 2), derived in the binary calibration setting, for each $h_l$ separately. Gupta and Ramdas (2021) proved a more general result for general $\ell_p$-ECE bounds. Similar results can also be derived for the suitably defined $\ell_p$-CW-ECE.

As discussed in Section 3.2, unlike previous works (Zadrozny and Elkan, 2002; Guo et al., 2017; Kull et al., 2019), Algorithm 9 does not normalize the $h_l$'s. We do not know how to derive Theorem 2 style results for a normalized version of Algorithm 9.

## D FIGURES FOR APPENDIX E

Appendix E begins on page 23. The relevant figures for Appendix E are displayed on the following pages.

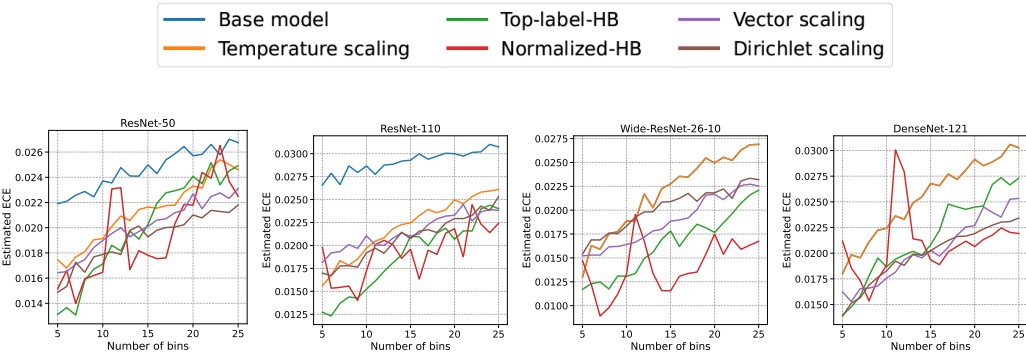

(a) TL-ECE estimates on CIFAR-10 with Brier score. TL-HB is close to the best in each case. While CW-HB performs the best at $B = 15$, the ECE estimate may not be reliable since it is highly variable across bins.

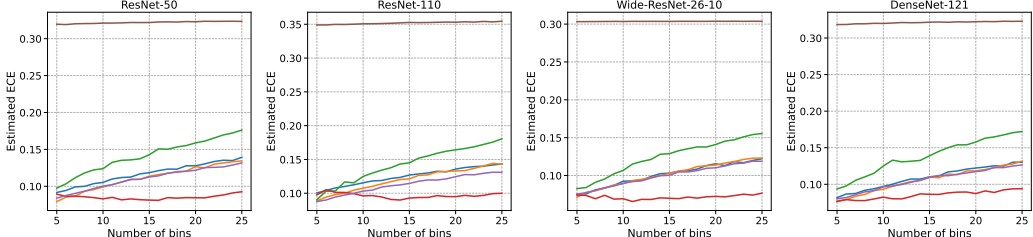

(b) TL-ECE estimates on CIFAR-100 with Brier score. N-HB is the best performing method, while DS is the worst performing method, across different numbers of bins. TL-HB performs worse than TS and VS.

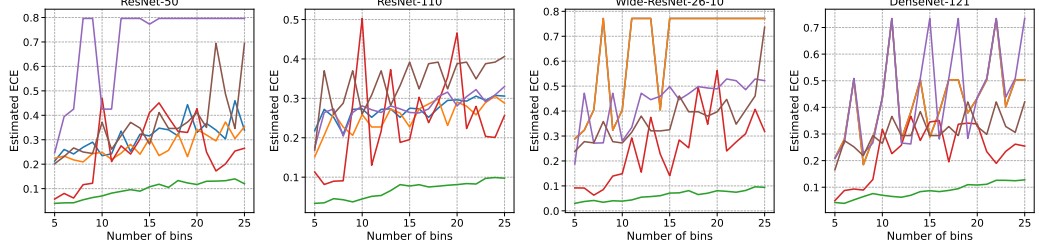

(c) TL-MCE estimates on CIFAR-10 with Brier score. The only reliably and consistently well-performing method is TL-HB.

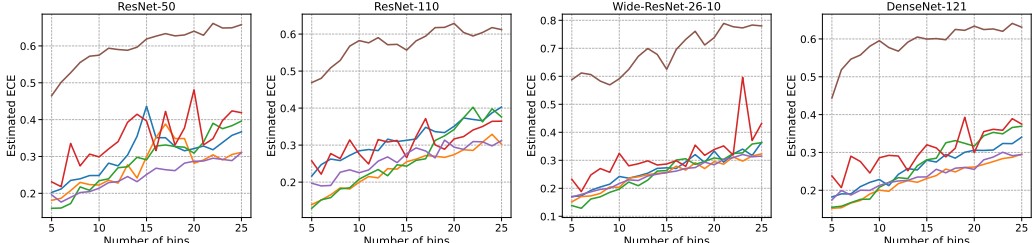

(d) TL-MCE estimates on CIFAR-100 with Brier score. DS is the worst performing method. Other methods perform across different values of $B$.

Figure 5: Table 2 style results with the number of bins varied as $B \in [5, 25]$. See Appendix E.5 for further details. The captions summarize the findings in each case. In most cases, the findings are similar to those with $B = 15$. The notable exception is that performance of N-HB on CIFAR-10 for TL-ECE while very good at $B = 15$, is quite inconsistent when seen across different bins. In some cases, the blue base model line and the orange temperature scaling line coincide. This occurs since the optimal temperature on the calibration data was learnt to be $T = 1$, which corresponds to not changing the base model at all.

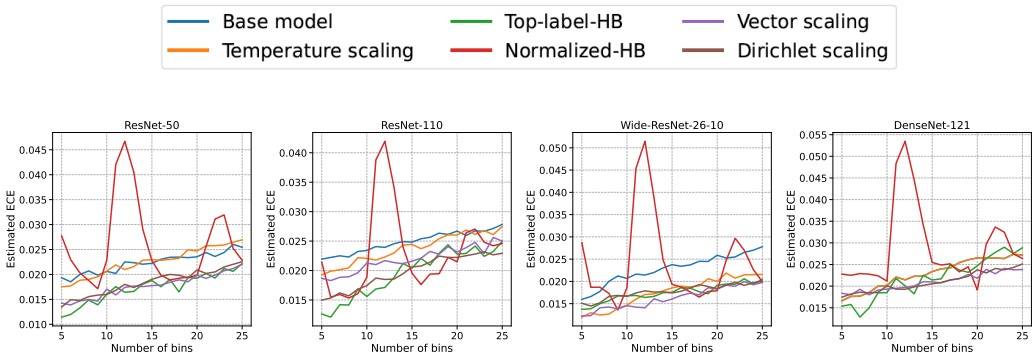

(a) TL-ECE estimates on CIFAR-10 with focal loss. TL-HB is close to the best in each case. While CW-HB performs the best at $B = 15$, the ECE estimate may not be reliable since it is highly variable across bins.

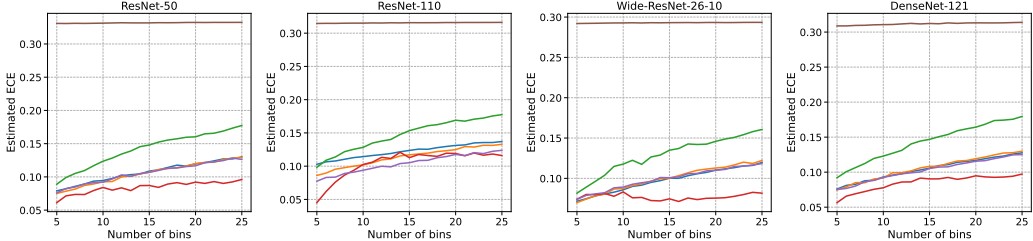

(b) TL-ECE estimates on CIFAR-100 with focal loss. N-HB is the best performing method, while DS is the worst performing method, across different numbers of bins. TL-HB performs worse than TS and VS.

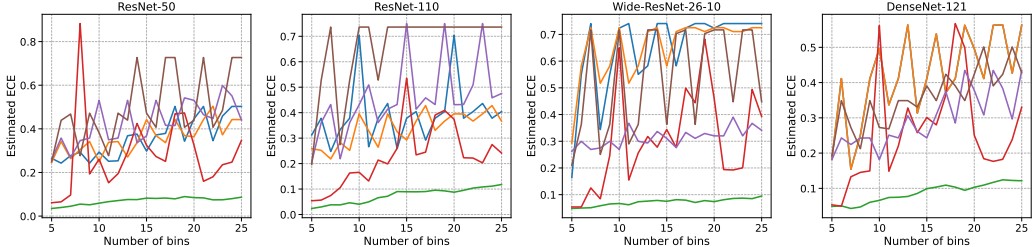

(c) TL-MCE estimates on CIFAR-10 with focal loss. The only reliably and consistently well-performing method is TL-HB.

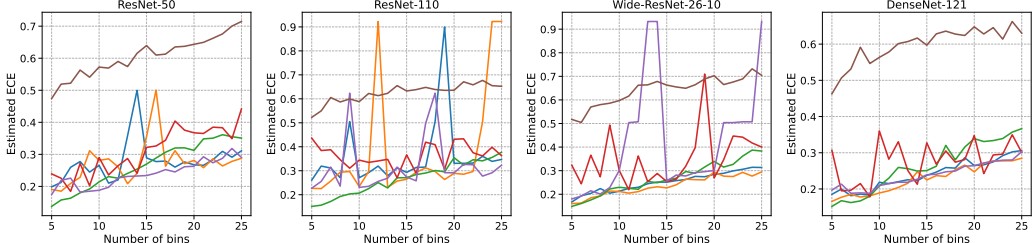

(d) TL-MCE estimates on CIFAR-100 with focal loss. DS is the worst performing method. Other methods perform across different values of $B$.

Figure 6: Table 4 style results with the number of bins varied as $B \in [5, 25]$. See Appendix E.5 for further details. The captions summarize the findings in each case. In most cases, the findings are similar to those with $B = 15$. In some cases, the blue base model line and the orange temperature scaling line coincide. This occurs since the optimal temperature on the calibration data was learnt to be $T = 1$, which corresponds to not changing the base model at all.

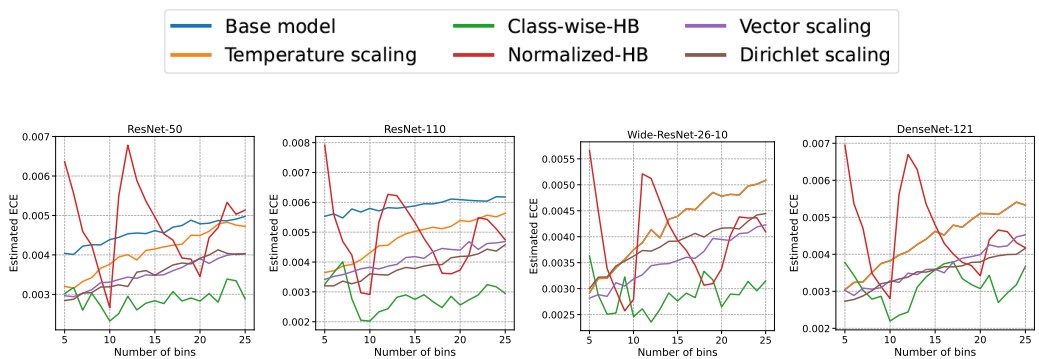

(a) CW-ECE estimates on CIFAR-10 with Brier score. CW-HB is the best performing method across bins, and N-HB is quite unreliable.

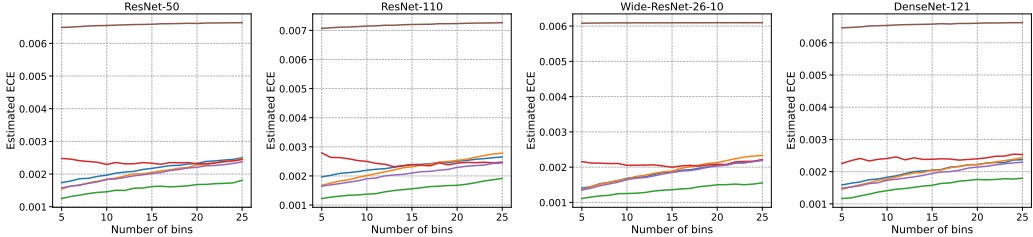

(b) CW-ECE estimates on CIFAR-100 with Brier score. CW-HB is the best performing method. DS and N-HB are the worst performing methods.

Figure 7: Table 3 style results with the number of bins varied as $B \in [5, 25]$. The captions summarize the findings in each case, which are consistent with those in the table. See Appendix E.5 for further details.

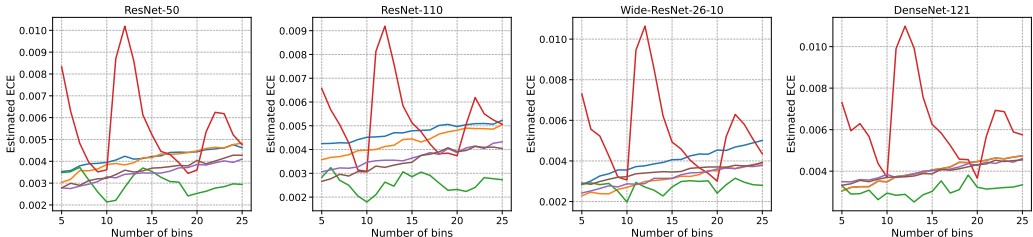

(a) CW-ECE estimates on CIFAR-10 with focal loss. CW-HB is the best performing method across bins, and N-HB is quite unreliable.

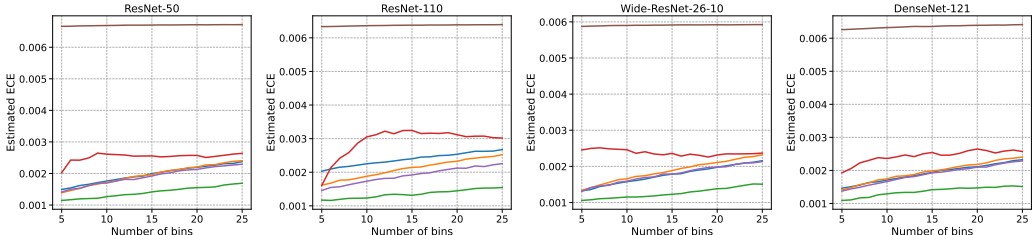

(b) CW-ECE estimates on CIFAR-100 with focal loss. CW-HB is the best performing method. DS and N-HB are the worst performing methods.

Figure 8: Table 5 style results with the number of bins varied as $B \in [5, 25]$. The captions summarize the findings in each case, which are consistent with those in the table. See Appendix E.5 for further details.

# E    ADDITIONAL EXPERIMENTAL DETAILS AND RESULTS FOR CIFAR-10 AND CIFAR-100

We present additional details and results to supplement the experiments with CIFAR-10 and CIFAR-100 in Sections 2 and 4 of the main paper.

## E.1    EXTERNAL LIBRARIES USED

All our base models were pre-trained deep-net models generated by Mukhoti et al. (2020), obtained from `www.robots.ox.ac.uk/~viveka/focal_calibration/` and used along with the code at `https://github.com/torrvision/focal_calibration` to obtain base predictions. We focused on the models trained with Brier score and focal loss, since it was found to perform the best for calibration. All reports in the main paper are with the Brier score; in Appendix E.4, we report corresponding results with focal loss.

We also used the code at `https://github.com/torrvision/focal_calibration` for temperature scaling (TS). For vector scaling (VS) and Dirichlet scaling (DS), we used the code of Kull et al. (2019), hosted at `https://github.com/dirichletcal/dirichlet_python`. For VS, we used the file `dirichletcal/calib/vectorscaling.py`, and for DS, we used the file `dirichletcal/calib/fulldirichlet.py`. No hyperparameter tuning was performed in any of our histogram binning experiments or baseline experiments; default settings were used in every case. The random seed was fixed so that every run of the experiment gives the same result. In particular, by relying on pre-trained models, we avoid training new deep-net models with multiple hyperparameters, thus avoiding any selection biases that may arise due to test-data peeking across multiple settings.

## E.2    FURTHER COMMENTS ON BINNING FOR ECE ESTIMATION

As mentioned in Remark 1, ECE estimates for all methods except TL-HB and CW-HB was done using fixed-width bins $[0, 1/B), [1/B, 2/B), \ldots [1 - 1/B, 1]$ for various values of $B \in [5, 25]$. For TL-HB and CW-HB, $B$ is the number of bins used for each call to binary HB. For TL-HB, note that we actually proposed that the number of bins-per-class should be fixed; see Section B.2. However, for ease of comparison to other methods, we simply set the number of bins to $B$ for each call to binary HB. That is, in line 5, we replace $\lfloor n_l/k \rfloor$ with $B$. For CW-HB, we described Algorithm 9 with different values of $k_l$ corresponding to the number of bins per class. For the CIFAR-10 and CIFAR-100 comparisons, we set each $k_1 = k_2 = \ldots = k_L = k$, where $k \in \mathbb{N}$ satisfies $\lfloor n/k \rfloor = B$.

Tables 2,3, 4, and 5 report estimates with $B = 15$, which has been commonly used in many works (Guo et al., 2017; Kull et al., 2019; Mukhoti et al., 2020). Corresponding to each table, we have a figure where ECE estimates with varying $B$ are reported to strengthen conclusions: these are Figure 5,7, 6, and 8 respectively. Plugin estimates of the ECE were used, same as Guo et al. (2017). Further binning was not done for TL-HB and CW-HB since the output is already discrete and sufficiently many points take each of the predicted values. Note that due to Jensen's inequality, any further binning will only decrease the ECE estimate (Kumar et al., 2019). Thus, using unbinned estimates may give TL-HB and CW-HB a disadvantage.

## E.3    SOME REMARKS ON MAXIMUM-CALIBRATION-ERROR (MCE)

Guo et al. (2017) defined MCE with respect to confidence calibration, as follows:

$$\text{conf-MCE}(c, h) := \sup_{r \in \text{Range}(h)} |P(Y = c(X) \mid h(X) = r) - r|. \tag{16}$$

Conf-MCE suffers from the same issue illustrated in Figure 2 for conf-ECE. In Figure 1b, we looked at the reliability diagram within two bins. These indicate two of the values over which the supremum is taken in equation (16): these are the Y-axis distances between the ★ markers and the $X = Y$ line for bins 6 and 10 (both are less than $0.02$). On the other hand, the effective *maximum* miscalibration for bin 6 is roughly $0.15$ (for class 1), and roughly $0.045$ (for class 4), and the maximum should be taken with respect to these values across all bins. To remedy the underestimation of the effective

| Metric | Dataset | Architecture | Base | TS | VS | DS | N-HB | TL-HB |
|---|---|---|---|---|---|---|---|---|
| Top-label-ECE | CIFAR-10 | ResNet-50 | 0.022 | 0.023 | **0.018** | 0.019 | 0.023 | 0.019 |
| | | ResNet-110 | 0.025 | 0.024 | 0.022 | 0.021 | **0.020** | **0.020** |
| | | WRN-26-10 | 0.024 | 0.019 | **0.016** | 0.017 | 0.019 | 0.018 |
| | | DenseNet-121 | 0.023 | 0.023 | **0.021** | 0.021 | 0.025 | **0.021** |
| | CIFAR-100 | ResNet-50 | 0.109 | 0.107 | 0.107 | 0.332 | **0.086** | 0.148 |
| | | ResNet-110 | 0.124 | 0.117 | **0.105** | 0.316 | 0.115 | 0.153 |
| | | WRN-26-10 | 0.100 | 0.100 | 0.101 | 0.293 | **0.074** | 0.135 |
| | | DenseNet-121 | 0.106 | 0.108 | 0.105 | 0.312 | **0.091** | 0.147 |
| Top-label-MCE | CIFAR-10 | ResNet-50 | 0.298 | 0.443 | 0.368 | 0.472 | 0.325 | **0.082** |
| | | ResNet-110 | 0.378 | 0.293 | 0.750 | 0.736 | 0.535 | **0.089** |
| | | WRN-26-10 | 0.741 | 0.582 | 0.311 | 0.363 | 0.344 | **0.075** |
| | | DenseNet-121 | 0.411 | 0.411 | 0.243 | 0.391 | 0.301 | **0.099** |
| | CIFAR-100 | ResNet-50 | 0.289 | 0.355 | **0.234** | 0.640 | 0.322 | 0.273 |
| | | ResNet-110 | 0.293 | **0.265** | 0.274 | 0.633 | 0.366 | 0.272 |
| | | WRN-26-10 | 0.251 | **0.227** | 0.256 | 0.663 | 0.229 | 0.270 |
| | | DenseNet-121 | 0.237 | **0.225** | 0.239 | 0.597 | 0.327 | 0.248 |

Table 4: Top-label-ECE and top-label-MCE for deep-net models and various post-hoc calibrators. All methods are same as Table 2. Best performing method in each row is in **bold**.

MCE, we can consider the top-label-MCE, defined as

$$\text{TL-MCE}(c, h) := \max_{l \in [L]} \sup_{r \in \text{Range}(h)} |P(Y = l \mid c(X) = l, h(X) = r) - r|. \tag{17}$$

Interpreted in words, the TL-MCE assesses the maximum deviation between the predicted and true probabilities across all predictions and all classes. Following the same argument as in the proof of Proposition 4, it can be shown that for any $c, h$, conf-MCE$(c, h) \leqslant$ TL-MCE$(c, h)$. The TL-MCE is closely related to conditional top-label calibration (Definition 1b). Clearly, an algorithm is $(\varepsilon, \alpha)$-conditionally top-label calibrated if and only if for every distribution $P$, $P(\text{TL-MCE}(c, h) \leqslant \varepsilon) \geqslant 1 - \alpha$. Thus the conditional top-label calibration guarantee of Theorem 1 implies a high probability bound on the TL-MCE as well.

### E.4 TABLE 2 AND 3 STYLE RESULTS WITH FOCAL LOSS

Results for top-label-ECE and top-label-MCE with the base deep net model being trained using focal loss are reported in Table 4. Corresponding results for class-wise-ECE are reported in Table 5. The observations are similar to the ones reported for Brier score:

1. For TL-ECE, TL-HB is either the best or close to the best performing method on CIFAR-10, but suffers on CIFAR-100. This phenomenon is discussed further in Appendix E.6. N-HB is the best or close to the best for both CIFAR-10 and CIFAR-100.

2. For TL-MCE, TL-HB is the best performing method on CIFAR-10, by a huge margin. For CIFAR-100, TS or VS perform better than TL-HB, but not by a huge margin.

3. For CW-ECE, CW-HB is the best performing method across the two datasets and all four architectures.

### E.5 ECE AND MCE ESTIMATES WITH VARYING NUMBER OF BINS

Corresponding to each entry in Tables 2 and 4, we perform an ablation study with the number of bins varying as $B \in [5, 25]$. This is in keeping with the findings of Roelofs et al. (2020) that the ECE/MCE estimate can vary with different numbers of bins, along with the relative performance of the various models.

The results are reported in Figure 5 (ablation of Table 2) and Figure 7 (ablation of Table 3). The captions of these figures contain further details on the findings. Most findings are similar to those in the main paper, but the findings in the tables are strengthened through this ablation. The same ablations are performed for focal loss as well. The results are reported in Figure 6 (ablation of

| Metric | Dataset | Architecture | Base | TS | VS | DS | N-HB | CW-HB |
|--------|---------|--------------|------|-----|-----|-----|------|-------|
| Class-wise-ECE $\times 10^2$ | CIFAR-10 | ResNet-50 | 0.42 | 0.42 | **0.35** | 0.37 | 0.52 | **0.35** |
| | | ResNet-110 | 0.48 | 0.44 | 0.36 | 0.35 | 0.51 | **0.29** |
| | | WRN-26-10 | 0.41 | 0.31 | 0.31 | 0.35 | 0.49 | **0.27** |
| | | DenseNet-121 | 0.41 | 0.41 | 0.40 | 0.39 | 0.63 | **0.30** |
| | CIFAR-100 | ResNet-50 | 0.22 | 0.20 | 0.20 | 0.66 | 0.23 | **0.16** |
| | | ResNet-110 | 0.24 | 0.23 | 0.21 | 0.72 | 0.24 | **0.16** |
| | | WRN-26-10 | 0.19 | 0.19 | 0.18 | 0.61 | 0.20 | **0.14** |
| | | DenseNet-121 | 0.20 | 0.21 | 0.19 | 0.66 | 0.24 | **0.16** |

Table 5: Class-wise-ECE for deep-net models and various post-hoc calibrators. All methods are same as Table 2, except top-label-HB is replaced with class-wise-HB or Algorithm 3 (CW-HB). Best performing method in each row is in **bold**.

Table 4) and Figure 8 (ablation of Table 5). The captions of these figures contain further details on the findings. The ablation results in the figures support those in the tables.

### E.6  ANALYZING THE POOR PERFORMANCE OF TL-HB ON CIFAR-100

CIFAR-100 is an imbalanced dataset with 100 classes and 5000 points for validation/calibration (as per the default splits). Due to random subsampling, the validation split we used had one of the classes predicted as the top-label only 31 times. Thus, based on Theorem 1, we do not expect HB to have small TL-ECE. This is confirmed by the empirical results presented in Tables 2/4, and Figures 5b/6b. We observe that HB has higher estimated TL-ECE than all methods except DS, for most values of the number of bins. The performance of TL-HB for TL-MCE however is much much closer to the other methods since HB uses the same number of points per bin, ensuring that the predictions are somewhat equally calibrated across bins (Figures 5d/6d). In comparison, for CW-ECE, CW-HB is the best performing method. This is because in the class-wise setting, 5000 points are available for recalibration irrespective of the class, which is sufficient for HB.

The deterioration in performance of HB when few calibration points are available was also observed in the binary setting by Gupta and Ramdas (2021, Appendix C). Niculescu-Mizil and Caruana (2005) noted in the conclusion of their paper that Platt scaling (Platt, 1999), which is closely related to TS, performs well when the data is small, but another nonparametric binning method, isotonic regression (Zadrozny and Elkan, 2002) performs better when enough data is available. Kull et al. (2019, Section 4.1) compared HB to other calibration techniques for class-wise calibration on 21 UCI datasets, and found that HB performs the worst. On inspecting the UCI repository, we found that most of the datasets they used had fewer than 5000 (total) data points, and many contain fewer than 500.

Overall, comparing our results to previous empirical studies, we believe that if sufficiently many points are available for recalibration, or the number of classes is small, then HB performs quite well. To be more precise, we expect HB to be competitive if at least 200 points per class can be held out for recalibration, and the number of points per bin is at least $k \geqslant 20$.

## F  ADDITIONAL EXPERIMENTAL DETAILS AND RESULTS FOR COVTYPE-7

We present additional details and results for the top-label HB experiment of Section B.2. The base classifier is an RF learnt using `sklearn.ensemble import RandomForestClassifier` with default parameters. The base RF is a nearly continuous base model since most predictions are unique. Thus, we need to use binning to make reliability diagrams, validity plots, and perform ECE estimation, for the base model. To have a fair comparison, instead of having a fixed binning scheme to assess the base model, the binning scheme was decided based on the unique predictions of top-label HB. Thus for every $l$, and $r \in \text{Range}(h_l)$, the bins are defined as $\{x : c(x) = l, h_l(x) = r\}$. Due to this, while the base model in Figures 4a and 4b are the same, the reliability diagrams and validity plots in orange are different. As can be seen in the bar plots in Figure 4, the ECE estimation is not affected significantly.

When $k = 100$, the total number of bins chosen by Algorithm 8 was 403, which is roughly 57.6 bins per class. The choice of $B = 50$ for the fixed bins per class experiment was made on this basis.

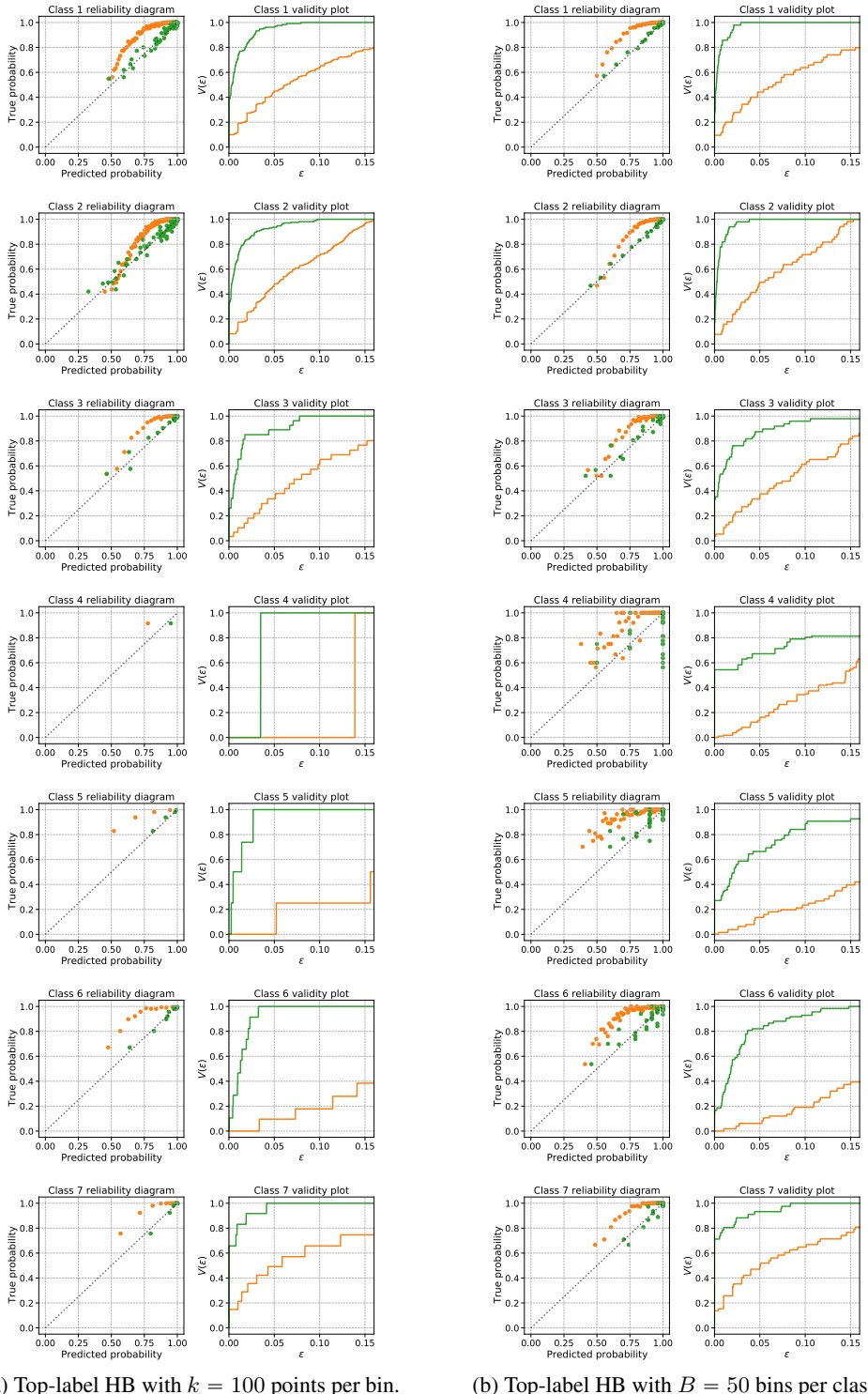

(a) Top-label HB with $k = 100$ points per bin.     (b) Top-label HB with $B = 50$ bins per class.

Figure 9: Top-label histogram binning (HB) calibrates a miscalibrated random-forest on the class imbalanced COVTYPE-7 dataset. For the less likely classes (4, 5, and 6), the left column is better calibrated than the right column. Similar observations are made on other datasets, and so we recommend adaptively choosing a different number of bins per class, as Algorithm 8 does.

Figure 9 supplements Figure 4 in the main paper by presenting reliability diagrams and validity plots of top-label HB for all classes. Figure 9a presents the plots with adaptive number of bins per class (Algorithm 8), and Figure 9b presents these for fixed number of bins per class. We make the following observations.

(a) For every class $l \in [L]$, the RF is overconfident. This may seem surprising at first since we generally expect that models may be overconfident for certain classes and underconfident for others. However, note that all our plots assess top-label calibration, that is, we are assessing the predicted and true probabilities of only the predicted class. It is possible that a model is overconfident for every class whenever that class is predicted to be the top-label.

(b) For the most likely classes, namely classes 1 and 2, the number of bins in the adaptive case is higher than 50. Fewer bins leads to better calibration (at the cost of sharpness). This can be verified through the validity plots for classes 1 and 2—the validity plots in the fixed bins case is slightly *above* the validity plot in the adaptive bin case. However both validity plots are quite similar.

(c) The opposite is true for the least likely classes, namely classes 4, 5, 6. The validity plot in the fixed bins case is *below* the validity plot in the adaptive bins case, indicating higher TL-ECE in the fixed bins case. The difference between the validity plots is high. Thus if a fixed number of bins per class is pre-decided, the performance for the least likely classes significantly suffers.

Based on these observations, we recommend adaptively choosing the number of bins per class, as done by Algorithm 8.

## G    BINNING-BASED CALIBRATORS FOR CANONICAL MULTICLASS CALIBRATION

Canonical calibration is a notion of calibration that does not fall in the M2B category. To define canonical calibration, we use $\mathbf{Y}$ to denote the output as a 1-hot vector. That is, $\mathbf{Y}_i = \mathbf{e}_{Y_i} \in \Delta^{L-1}$, where $e_l$ corresponds to the $l$-th canonical basis vector in $\mathbb{R}^d$. Recall that a predictor $\mathbf{h} = (h_1, h_2, \ldots, h_L)$ is said to be canonically calibrated if $P(Y = l \mid \mathbf{h}(X)) = h_l(X)$ for every $l \in [L]$. Equivalently, this can be stated as $\mathbb{E}\left[\mathbf{Y} \mid \mathbf{h}(X)\right] = \mathbf{h}(X)$. Canonical calibration implies class-wise calibration:

**Proposition 1.** *If* $\mathbb{E}\left[\mathbf{Y} \mid \mathbf{h}(X)\right] = \mathbf{h}(X)$*, then for every* $l \in [L]$*,* $P(Y = l \mid h_l(X)) = h_l(X)$*.*

The proof in Appendix H is straightforward, but the statement above is illuminating, because there exist predictors that are class-wise calibrated but not canonically calibrated (Vaicenavicius et al., 2019, Example 1).

Canonical calibration is not an M2B notion since the conditioning occurs on the $L$-dimensional prediction vector $\text{pred}(X) = \mathbf{h}(X)$, and after this conditioning, each of the $L$ statements $P(Y = l \mid \text{pred}(X)) = h_l(X)$ should *simultaneously* be true. On the other hand, M2B notions verify only individual binary calibration claims for every such conditioning. Since canonical calibration does not fall in the M2B category, Algorithm 5 does not lead to a calibrator for canonical calibration. In this section, we discuss alternative binning-based approaches to achieving canonical calibration.

For binary calibration, there is a complete ordering on the interval $[0, 1]$, and this ordering is leveraged by binning based calibration algorithms. However, $\Delta^{L-1}$, for $L \geqslant 3$ does not have such a natural ordering. Hence, binning algorithms do not obviously extend for multiclass classification. In this section, we briefly discuss some binning-based calibrators for canonical calibration. Our descriptions are for general $L \geqslant 3$, but we anticipate these algorithms to work reasonably only for small $L$, say if $L \leqslant 5$.

As usual, denote $\mathbf{g} : \mathcal{X} \to \Delta^{L-1}$ as the base model and $\mathbf{h} : \mathcal{X} \to \Delta^{L-1}$ as the model learnt using some post-hoc canonical calibrator. For canonical calibration, we can surmise binning schemes that directly learn $\mathbf{h}$ by partitioning the prediction space $\Delta^{L-1}$ into bins and estimating the distribution of $\mathbf{Y}$ in each bin. A canonical calibration guarantee can be showed for such a binning scheme using multinomial concentration (Podkopaev and Ramdas, 2021, Section 3.1). However, since $\text{Vol}(\Delta^{L-1}) = 2^{\Theta(L)}$, there will either be a bin whose volume is $2^{\Omega(L)}$ (meaning that $\mathbf{h}$ would

not be sharp), or the number of bins will be $2^{\Omega(L)}$, entailing $2^{\Omega(L)}$ requirements on the sample complexity—a *curse of dimensionality*. Nevertheless, let us consider some binning schemes that could work if $L$ is small.

Formally, a binning scheme corresponds to a partitioning of $\Delta^{L-1}$ into $B \geqslant 1$ bins. We denote this binning scheme as $\mathcal{B} : \Delta^{L-1} \to [B]$, where $\mathcal{B}(\mathbf{s})$ corresponds to the bin to which $\mathbf{s} \in \Delta^{L-1}$ belongs. To learn $\mathbf{h}$, the calibration data is binned to get sets of data-point indices that belong to each bin, depending on the $\mathbf{g}(X_i)$ values:

$$\text{for every } b \in [B], \; T_b := \{i : \mathcal{B}(\mathbf{g}(X_i)) = b\}, n_b = |T_b|.$$

We then compute the following estimates for the label probabilities in each bin:

$$\text{for every } (l,b) \in [L] \times [B], \; \widehat{\Pi}_{l,b} := \frac{\sum_{i \in T_b} \mathbb{1}\{Y_i = l\}}{n_b} \text{ if } n_b > 0 \text{ else } \widehat{\Pi}_{l,b} = 1/B.$$

The binning predictor $\mathbf{h} : \mathcal{X} \to \Delta^{L-1}$ is now defined component-wise as follows:

$$\text{for every } l \in [L], \; h_l(x) = \widehat{\Pi}_{l,\mathcal{B}(x)}.$$

In words, for every bin $b \in [B]$, $\mathbf{h}$ predicts the empirical distribution of the $Y$ values in bin $b$.

Using a multinomial concentration inequality (Devroye, 1983; Qian et al., 2020; Weissman et al., 2003), calibration guarantees can be shown for the learnt $\mathbf{h}$. Podkopaev and Ramdas (2021, Theorem 3) show such a result using the Bretagnolle-Huber-Carol inequality. All of these concentration inequality give bounds that depend inversely on $n_b$ or $\sqrt{n_b}$.

In the following subsections, we describe some binning schemes which can be used for canonical calibration based on the setup illustrated above. First we describe fixed schemes that are not adaptive to the distribution of the data: Sierpinski binning (Appendix G.1) and grid-style binning (Appendix G.2). These are analogous to fixed-width binning for $L = 2$. Fixed binning schemes are not adapted to the calibration data and may have highly imbalanced bins leading to poor estimation of the distribution of $\mathbf{Y}$ in bins with small $n_b$. In the binary case, this issue is remedied using histogram binning to ensure that each bin has nearly the same number of calibration points (Gupta and Ramdas, 2021). While histogram binning uses the order of the scalar $g(X_i)$ values, there is no obvious ordering for the multi-dimensional $\mathbf{g}(X_i)$ values. In Appendix G.3 we describe a projection based histogram binning scheme that circumvents this issue and ensures that each $n_b$ is reasonably large. In Appendix G.4, we present some preliminary experimental results using our proposed binning schemes.

Certain asymptotic consistency results different from calibration have been established for histogram regression and classification in the nonparametric statistics literature (Nobel, 1996; Lugosi and Nobel, 1996; Gordon and Olshen, 1984; Breiman et al., 2017; Devroye, 1988); further extensive references can be found within these works. The methodology of histogram regression and classification relies on binning and is very similar to the one we propose here. The main difference is that these works consider binning the feature space $\mathcal{X}$ directly, unlike the post-hoc setting where we are essentially interested in binning $\Delta^{L-1}$. In terms of theory, the results these works target are asymptotic consistency for the (Bayes) optimal classification and regression functions, instead of canonical calibration. It would be interesting to consider the (finite-sample) canonical calibration properties of the various algorithms proposed in the context of histogram classification. However, such a study is beyond the scope of this paper.

### G.1 SIERPINSKI BINNING

First, we describe Sierpinski binning for $L = 3$. The probability simplex for $L = 3$, $\Delta^2$, is a triangle with vertices $\mathbf{e}_1 = (1,0,0)$, $\mathbf{e}_2 = (0,1,0)$, and $\mathbf{e}_3 = (0,0,1)$. Sierpinski binning is a recursive partitioning of this triangle based on the fractal popularly known as the Sierpinski triangles. Some Sierpinski bins for $L = 3$ are shown in Figure 10. Formally, we define Sierpinski binning recursively based on a depth parameter $q \in \mathbb{N}$. Given an $x \in \mathcal{X}$, let $\mathbf{s} = \mathbf{g}(x)$. For $q = 1$, the number of bins is $B = 4$, and the binning scheme $\mathcal{B}$ is defined as:

$$\mathcal{B}(\mathbf{s}) = \begin{cases} 1 & \text{if } s_1 > 0.5 \\ 2 & \text{if } s_2 > 0.5 \\ 3 & \text{if } s_3 > 0.5 \\ 4 & \text{otherwise.} \end{cases} \tag{18}$$

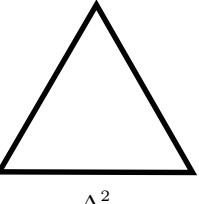 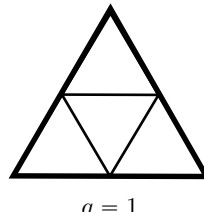 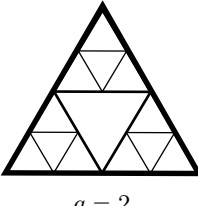 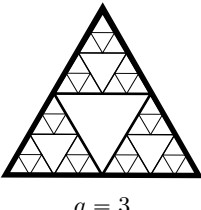

$\Delta^2$      $q = 1$      $q = 2$      $q = 3$

Figure 10: Sierpinski binning for $L = 3$. The leftmost triangle represents the probability simplex $\Delta^2$. Sierpinski binning divides $\Delta^2$ recursively based on a depth parameter $q \in \mathbb{N}$.

Note that since $s_1 + s_2 + s_3 = 1$, only one of the above conditions can be true. It can be verified that each of the bins have volume equal to $(1/4)$-th the volume of the probability simplex $\Delta^2$. If a finer resolution of $\Delta^2$ is desired, $B$ can be increased by further dividing the partitions above. Note that each partition is itself a triangle; thus each triangle can be mapped to $\Delta^2$ to recursively define the sub-partitioning. For $i \in [4]$, define the bins $b_i = \{\mathbf{s} : \mathcal{B}(\mathbf{s}) = i\}$. Consider the bin $b_1$. Let us *reparameterize* it as $(t_1, t_2, t_3) = (2s_1 - 1, 2s_2, 2s_3)$. It can be verified that

$$b_1 = \{(t_1, t_2, t_3) : s_1 > 0.5\} = \{(t_1, t_2, t_3) : t_1 + t_2 + t_3 = 1, t_1 \in (0, 1], t_2 \in [0, 1), t_3 \in [0, 1)\}.$$

Based on this reparameterization, we can recursively sub-partition $b_1$ as per the scheme (18), replacing $s$ with $t$. Such reparameterizations can be defined for each of the bins defined in (18):

$$b_2 = \{(s_1, s_2, s_3) : s_2 > 0.5\} : (t_1, t_2, t_3) = (2s_1, 2s_2 - 1, 2s_3),$$
$$b_3 = \{(s_1, s_2, s_3) : s_3 > 0.5\} : (t_1, t_2, t_3) = (2s_1, 2s_2, 2s_3 - 1),$$
$$b_4 = \{(s_1, s_2, s_3) : s_i \leqslant 0.5 \text{ for all } i\} : (t_1, t_2, t_3) = (1 - 2s_1, 1 - 2s_2, 1 - 2s_3),$$

and thus every bin can be recursively sub-partitioned as per (18). As illustrated in Figure 10, for Sierpinski binning, we sub-partition only the bins $b_1, b_2, b_3$ since the bin $b_4$ corresponds to low confidence for all labels, where finer calibration may not be needed. (Also, in the $L > 3$ case described shortly, the corresponding version of $b_4$ is geometrically different from $\Delta^{L-1}$, and the recursive partitioning cannot be defined for it.) If at every depth, we sub-partition all bins except the corresponding $b_4$ bins, then it can be shown using simple algebra that the total number of bins is $(3^{q+1} - 1)/2$. For example, in Figure 10, when $q = 2$, the number of bins is $B = 14$, and when $q = 3$, the number of bins is $B = 40$.

As in the case of $L = 3$, Sierpinski binning for general $L$ is defined through a partitioning function of $\Delta^{L-1}$ into $L + 1$ bins, and a reparameterization of the partitions so that they can be further sub-partitioned. The $L + 1$ bins at depth $q = 1$ are defined as

$$\mathcal{B}(\mathbf{s}) = \begin{cases} l & \text{if } s_l > 0.5, \\ L + 1 & \text{otherwise.} \end{cases} \tag{19}$$

While this looks similar to the partitioning (18), the main difference is that the bin $b_{L+1}$ has a larger volume than other bins. Indeed for $l \in [L]$, $\text{vol}(b_l) = \text{vol}(\Delta^{L-1})/2^{L-1}$, while $\text{vol}(b_{L+1}) = \text{vol}(\Delta^{L-1})(1 - L/2^{L-1}) \geqslant \text{vol}(\Delta^{L-1})/2^{L-1}$, with equality only occuring for $L = 3$. Thus the bin $b_{L+1}$ is larger than the other bins. If $\mathbf{g}(x) \in b_{L+1}$, then the prediction for $x$ may be not be very sharp, compared to if $\mathbf{g}(x)$ were in any of the other bins. On the other hand, if $\mathbf{g}(x) \in b_{L+1}$, the score for every class is smaller than $0.5$, and sharp calibration may often not be desired in this region.

In keeping with this understanding, we only reparameterize the bins $b_1, b_2, \ldots, b_L$ so that they can be further divided:

$$b_l = \{(s_1, s_2, \ldots, s_L) : s_l > 0.5\} : (t_1, t_2, \ldots, t_L) = (2s_1, \ldots, 2s_l - 1, \ldots, 2s_L).$$

For every $l \in [L]$, under the reparameterization above, it is straightforward to verify that

$$\{(t_1, t_2, \ldots, t_L) : s_l > 0.5\} = \{(t_1, t_2, \ldots, t_L) : \sum_{u \in [L]} t_u = 1, t_l \in (0, 1], t_u \in [0, 1) \, \forall u \neq l\}.$$

Thus every bin can be recursively sub-partitioned following (19). For Sierpinski binning with $L$ labels, the number of bins at depth $q$ is given by $(L^{q+1} - 1)/(L - 1)$.

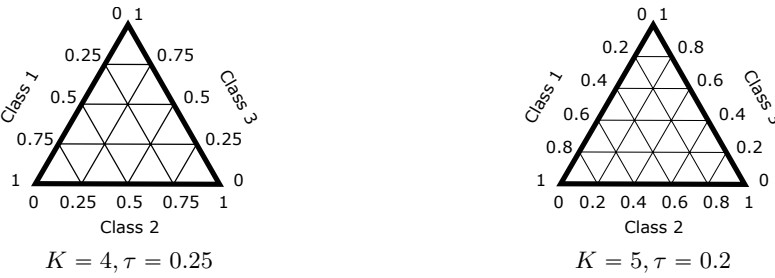

Figure 11: Grid-style binning for $L = 3$.

## G.2 GRID-STYLE BINNING

Grid-style binning is motivated from the 2D reliability diagrams of Widmann et al. (2019, Figure 1), where they partitioned $\Delta^2$ into multiple equi-volume bins in order to assess canonical calibration on a 3-class version of CIFAR-10. For $L = 3$, $\Delta^2$ can be divided as shown in Figure 11. This corresponds to *gridding* the space $\Delta^2$, just the way we think of *gridding* the real hyperplane. However, the mathematical description of this grid for general $L$ is not apparent from Figure 11. We describe grid-style binning formally for general $L \geqslant 3$.

Consider some $\tau > 0$ such that $K := 1/\tau \in \mathbb{N}$. For every tuple $\mathbf{k} = (k_1, k_2, \ldots, k_L)$ in the set

$$I = \{\mathbf{k} \in \mathbb{N}^L : \max(L, K + 1) \leqslant \sum_{l \in [L]} k_l \leqslant K + (L - 1)\}, \tag{20}$$

define the bins

$$b_{\mathbf{k}} := \{\mathbf{s} \in \Delta^{L-1} : \text{ for every } l \in [L], s_l K \in [k_l - 1, k_l]. \tag{21}$$

These bins are not mutually disjoint, but intersections can only occur at the edges. That is, for every $\mathbf{s}$ that belongs to more than one bin, at least one component $s_l$ satisfies $s_l K \in \mathbb{N}$. In order to identify a single bin $\mathbf{s}$, ties can be broken arbitrarily. One strategy is to use some ordering on $\mathbb{N}^L$; say for $\mathbf{k}_1 \neq \mathbf{k}_2 \in \mathbb{N}$, $\mathbf{k}_1 < \mathbf{k}_2$ if and only if for the first element of $\mathbf{k}_1$ that is unequal to the corresponding element of $\mathbf{k}_2$ the one corresponding to $\mathbf{k}_1$ is smaller. Then define the binning function $\mathcal{B} : \Delta^{L-1} \rightarrow |I|$ as $\mathcal{B}(\mathbf{s}) = \min\{\mathbf{k} : \mathbf{s} \in b_{\mathbf{k}}\}$. The following propositions prove that a) each $\mathbf{s}$ belongs to at least one bin, and b) that every bin is an $L - 1$ dimensional object (and thus a meaningful partition of $\Delta^{L-1}$).

**Proposition 2.** *The bins $\{b_{\mathbf{k}} : \mathbf{k} \in I\}$ defined by (21) mutually exhaust $\Delta^{L-1}$.*

*Proof.* Consider any $\mathbf{s} \in \Delta^{L-1}$. For $s_l K \notin \mathbb{N} = \{1, 2, \ldots\}$, set $k_l = \max(1, \lceil s_l K \rceil) > s_l K$. Consider the condition

$$C : \text{ for all } l \text{ such that } s_l K \notin \mathbb{N}, s_l K = 0.$$

If $C$ is true, then for $l$ such that $s_l K \in \mathbb{N}$, set $k_l = s_l K$. If $C$ is not true, then for $l$ such that $s_l K \in \mathbb{N}$, set exactly one $k_l = s_l K + 1$, and for the rest set $k_l = s_l K$. Based on this setting of $\mathbf{k}$, it can be verified that $\mathbf{s} \in b_{\mathbf{k}}$.

Further, note that for every $l$, $k_l \geqslant s_l K$, and there exists at least one $l$ such that $k_l > s_l K$. Thus we have:

$$\sum_{l=1}^{L} k_l > \sum_{l=1}^{L} s_l K$$
$$= K.$$

Since $\sum_{l=1}^{L} k_l \in \mathbb{N}$, we must have $\sum_{l=1}^{L} k_l \geqslant K + 1$. Further since each $k_l \in \mathbb{N}$, $\sum_{l=1}^{L} k_l \geqslant L$.

Next, note that for every $l$, $k_l \leqslant s_l K + 1$. If $C$ is true, then there is at least one $l$ such that $s_l K \in \mathbb{N}$, and for this $l$, we have set $k_l = s_l K < s_l K + 1$. If $C$ is not true, then either there exists at least one $l$ such that $s_l K \notin \mathbb{N} \cup \{0\}$ for which $k_l = \lceil s_l K \rceil < s_l K + 1$, or every $s_l K \in \mathbb{N}$, in which case we

have set $k_l = s_l K$ for one of them. In all cases, note that there exists an $l$ such that $k_l < s_l K + 1$. Thus,

$$\sum_{l=1}^{L} k_l < \sum_{l=1}^{L} (s_l K + 1)$$
$$= K + L.$$

Since $\sum_{l=1}^{L} k_l \in \mathbb{N}$, we must have $\sum_{l=1}^{L} k_l \leqslant K + L - 1$.

$\square$

Next, we show that each bin indexed by $\mathbf{k} \in I$ contains a non-zero volume subset of $\Delta^{L-1}$, where volume is defined with respect to the Lebesgue measure in $\mathbb{R}^{L-1}$. This can be shown by arguing that $b_{\mathbf{k}}$ contains a scaled and translated version of $\Delta^{L=1}$.

**Proposition 3.** *For every $\mathbf{k} \in I$, there exists some $\mathbf{u} \in \mathbb{R}^L$ and $v > 0$ such that $\mathbf{u} + v\Delta^{L-1} \subseteq b_{\mathbf{k}}$.*

*Proof.* Fix some $\mathbf{k} \in I$. By condition (20), $\sum_{l=1}^{L} k_l \in [\max(L, K+1), K+L-1]$. Based on this, we claim that there exists a $\tau \in [0, 1)$ such that

$$\sum_{l=1}^{L} (k_l - 1) + \tau L + (1 - \tau) = K. \tag{22}$$

Indeed, note that for $\tau = 0$, $\sum_{l=1}^{L} (k_l - 1) + \tau L + (1 - \tau) \leqslant (K - 1) + 1 = K$ and for $\tau = 1$, $\sum_{l=1}^{L} (k_l - 1) + \tau L + (1 - \tau) = \sum_{l=1}^{L} k_l > K$. Thus, there exists a $\tau$ that satisfies (22) by the intermediate value theorem.

Next, define $\mathbf{u} = K^{-1}(\mathbf{k} + (\tau - 1)\mathbf{1}_L)$ and $v = K^{-1}(1 - \tau) > 0$, where $\mathbf{1}_L$ denotes the vector in $\mathbb{R}^L$ with each component equal to 1. Consider any $\mathbf{s} \in \mathbf{u} + v\Delta^{L-1}$. Note that for every $l \in [L]$, $s_l K \in [k_l - 1, k_l]$ and by property (22),

$$\sum_{l=1}^{L} s_l K = \left( \sum_{l=1}^{L} (k_l + (\tau - 1)) \right) + v = \sum_{l=1}^{L} (k_l - 1) + \tau L + (1 - \tau) = K.$$

Thus, $\mathbf{s} \in \Delta^{L-1}$ and by the definition of $b_{\mathbf{k}}$, $\mathbf{s} \in b_{\mathbf{k}}$. This completes the proof.

$\square$

The previous two propositions imply that $\mathcal{B}$ satisfies the property we require of a reasonable binning scheme. For $L = 3$, grid-style binning gives equi-volume bins as illustrated in Figure 11; however this is not true for $L > 3$. We now describe a histogram binning based partitioning scheme.

### G.3 Projection based histogram binning for canonical calibration

Some of the bins defined by Sierpinski binning and grid-style binning may have very few calibration points $n_b$, leading to poor estimation of $\widehat{\Pi}$. In the binary calibration case, this can be remedied using histogram binning which strongly relies on the scoring function $g$ taking values in a fully ordered space $[0, 1]$. To ensure that each bin contains $\Omega(n/B)$ points, we estimate the quantiles of $g(X)$ and created the bins as per these quantiles. However, there are no natural quantiles for unordered prediction spaces such as $\Delta^{L-1}$ ($L \geqslant 3$). In this section, we develop a histogram binning scheme for $\Delta^{L-1}$ that is semantically interpretable and has desirable statistical properties.

Our algorithm takes as input a prescribed number of bins $B$ and an arbitrary sequence of vectors $q_1, q_2, \ldots, q_{B-1} \in \mathbb{R}^L$ with unit $\ell_2$-norm: $\|q_i\|_2 = 1$. Each of these vectors represents a direction on which we will project $\Delta^{L-1}$ in order to induce a full order on $\Delta^{L-1}$. Then, for each direction, we will use an order statistics on the induced full order to identify a bin with exactly $\lfloor (n+1)/B \rfloor - 1$ calibration points (except the last bin, which may have more points). The formal algorithm is described in Algorithm 10. It uses the following new notation: given $m$ vectors $v_1, v_2, \ldots, v_m \in \mathbb{R}^L$, a unit vector $u$, and an index $j \in [m]$, let order-statistics($\{v_1, v_2, \ldots, v_m\}, u, j$) denote the $j$-th order-statistics of $\{v_1^T u, v_2^T u, \ldots, v_m^T u\}$.

---

**Algorithm 10:** Projection histogram binning for canonical calibration

---

**Input:** Base multiclass predictor $\mathbf{g} : \mathcal{X} \to \Delta^{L-1}$, calibration data
$\qquad \mathcal{D} = \{(X_1, Y_1), (X_2, Y_2), \ldots, (X_n, Y_n)\}$
**Hyperparameter:** number of bins $B$, unit vectors $q_1, q_2, \ldots, q_B \in \mathbb{R}^L$,
**Output:** Approximately calibrated scoring function $\mathbf{h}$

1   $S \leftarrow \{\mathbf{g}(X_1), \mathbf{g}(X_2), \ldots, \mathbf{g}(X_n)\}$;
2   $T \leftarrow$ empty array of size $B$;
3   $c \leftarrow \lfloor \frac{n+1}{B} \rfloor$;
4   **for** $b \leftarrow 1$ **to** $B - 1$ **do**
5      $T_b \leftarrow$ order-statistics$(S, q_b, c)$;
6      $S \leftarrow S \backslash \{v \in S : v^T q_b \leqslant T_b\}$;
7   **end**
8   $T_B \leftarrow 1.01$;
9   $\mathcal{B}(\mathbf{g}(\cdot)) \leftarrow \min\{b \in [B] : \mathbf{g}(\cdot)^T q_b < T_b\}$;
10   $\widehat{\Pi} \leftarrow$ empty matrix of size $B \times L$;
11   **for** $b \leftarrow 1$ **to** $B$ **do**
12      **for** $l \leftarrow 1$ **to** $L$ **do**
13         $\widehat{\Pi}_{b,l} \leftarrow \text{Mean}\{\mathbb{1}\{Y_i = l\} : \mathcal{B}(\mathbf{g}(X_i)) = b \text{ and } \forall s \in [B], \mathbf{g}(X_i)^T q_s \neq T_s\}$;
14      **end**
15   **end**
16   **for** $l \leftarrow 1$ **to** $L$ **do**
17      $h_l(\cdot) \leftarrow \widehat{\Pi}_{\mathcal{B}(\mathbf{g}(\cdot)),l}$;
18   **end**
19   **return** $\mathbf{h}$;

---

We now briefly describe some values computed by Algorithm 10 in words to build intuition. The array $T$, which is learnt on the data, represents the identified thresholds for the directions given by $q$. Each $(q_b, T_b)$ pair corresponds to a hyperplane that *cuts* $\Delta^{L-1}$ into two subsets given by $\{x \in \Delta^{L-1} : x^T q_b < T_b\}$ and $\{x \in \Delta^{L-1} : x^T q_b \geqslant T_b\}$. The overall partitioning of $\Delta^{L-1}$ is created by merging these cuts sequentially. This defines the binning function $\mathcal{B}$. By construction, the binning function is such that each bin contains at least $\lfloor \frac{n+1}{B} \rfloor - 1$ many points in its interior. As suggested by Gupta and Ramdas (2021), we do not include the points that lie on the boundary, that is, points $X_i$ that satisfy $\mathbf{g}(X_i)^T q_s = T_s$ for some $s \in [B]$. The interior points bins are then used to estimate the bin biases $\widehat{\Pi}$.

No matter how the $q$-vectors are chosen, the bins created by Algorithm 10 have at least $\lfloor \frac{n}{B} \rfloor - 1$ points for bias estimation. However, we discuss some simple heuristics for setting $q$ that are semantically meaningful. For some intuition, note that the binary version of histogram binning Gupta and Ramdas (2021, Algorithm 1) is essentially recovered by Algorithm 10 if $L = 2$ by setting each $q_b$ as $\mathbf{e}_2$ (the vector $[0, 1]$). Equivalently, we can set each $q_b$ to $-\mathbf{e}_1$ since both are equivalent for creating a projection-based order on $\Delta_2$. Thus for $L \geqslant 3$, a natural strategy for the $q$-vectors is to rotate between the canonical basis vectors: $q_1 = -\mathbf{e}_1, q_2 = -\mathbf{e}_2, \ldots, q_L = -\mathbf{e}_L, q_{L+1} = -\mathbf{e}_1, \ldots$, and so on. Projecting with respect to $-\mathbf{e}_l$ focuses on the class $l$ by forming a bin corresponding to the largest values of $g_l(X_i)$ among the remaining $X_i$'s which have not yet been binned. (On the other hand, projecting with respect to $\mathbf{e}_l$ will correspond to forming a bin with the smallest values of $g_l(X_i)$.)

The $q$-vectors can also be set adaptively based on the training data (without seeing the calibration data). For instance, if most points belong to a specific class $l \in [L]$, we may want more sharpness for this particular class. In that case, we can choose a higher ratio of the $q$-vectors to be $-\mathbf{e}_l$.

### G.4   EXPERIMENTS WITH THE COVTYPE DATASET

In Figure 12 we illustrate the binning schemes proposed in this section on a 3-class version of the COVTYPE-7 dataset considered in Section B.2. As noted before, this is an imbalanced dataset where classes 1 and 2 dominate. We created a 3 class problem with the classes 1, 2, and other (as

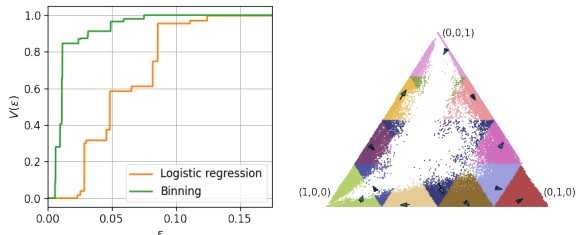

(a) Calibration using Sierpinski binning at depth $q = 2$.

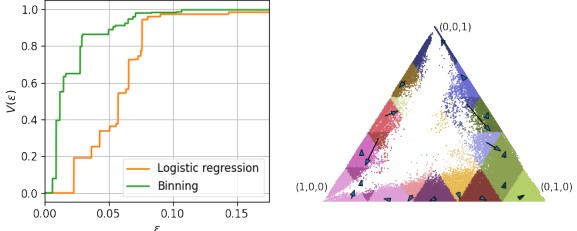

(b) Calibration using grid-style binning with $K = 5$, $\tau = 0.2$.

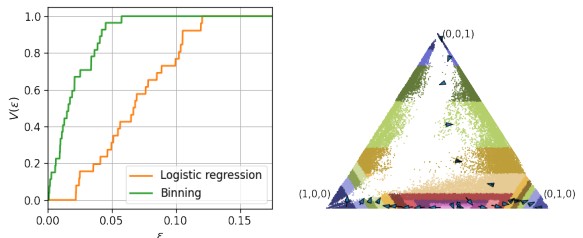

(c) Projection-based HB with $B = 27$ projections: $q_1 = -\mathbf{e}_1, q_2 = -\mathbf{e}_2, \ldots, q_4, -\mathbf{e}_1, \ldots$, and so on.

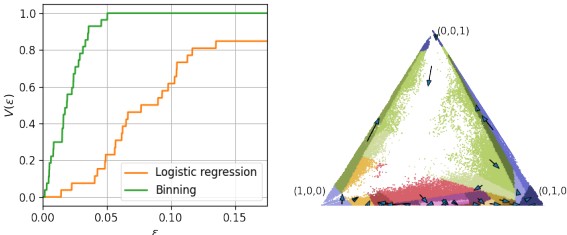

(d) Projection-based HB with $B = 27$ random projections ($q_i$ drawn uniformly from the $\ell_2$-unit-ball in $\mathbb{R}^3$).

Figure 12: Canonical calibration using fixed and histogram binning on a 3-class version of the COVTYPE-7 dataset. The reliability diagrams (left) indicate that all forms of binning improve the calibration of the base logistic regression model. The recalibration diagrams (right) are a scatter plot of the predictions $\mathbf{g}(X)$ on the test data with the points colored in 10 different colors depending on their bin. For every bin, the arrow-tail indicates the mean probability predicted by the base model $\mathbf{g}$ whereas the arrow-head indicates the probability predicted by the updated model $\mathbf{h}$.

class 3). The entire dataset has 581012 points and the ratio of the classes is approximately 36%, 49%, 15% respectively. The dataset was split into train-test in the ratio 70:30. The training data was further split into modeling-calibration in the ratio 90:10. A logistic regression model $\mathbf{g}$ using `sklearn.linear_model.LogisticRegression` was learnt on the modeling data, and $\mathbf{g}$ was recalibrated on the calibration data.

The plots on the right in Figure 12 are *recalibration diagrams*. The base predictions $\mathbf{g}(X)$ on the test-data are displayed as a scatter plot on $\Delta^2$. Points in different bins are colored using one of 10

different colors (since the number of bins is larger than 10, some colors correspond to more than one bin). For each bin, an arrow is drawn, where the tail of the arrow corresponds to the average $\mathbf{g}(X)$ predictions in the bin and the head of the arrow corresponds to the recalibrated $\mathbf{h}(X)$ prediction for the bin. For bins that contained very few points, the arrows are suppressed for visual clarity.

The plots on the left in Figure 12 are validity plots (Gupta and Ramdas, 2021). Validity plots display estimates of
$$V(\varepsilon) = P_{\text{test-data}}\left(\|\mathbb{E}\left[\mathbf{Y} \mid \mathbf{g}(X)\right] - \mathbf{g}(X)\|_1 \leqslant \varepsilon\right),$$
as $\varepsilon$ varies ($\mathbf{g}$ corresponds to the validity plot for logistic regression; replacing $\mathbf{g}$ with $\mathbf{h}$ above gives plots for the binning based classifier $\mathbf{h}$). For logistic regression, the same binning scheme as the one provided by $\mathbf{h}$ is used to estimate $V(\varepsilon)$.

Overall, Figure 12 shows that all of the binning approaches improve the calibration of the original logistic regression model across different $\varepsilon$. However, the recalibration does not change the original model significantly. Comparing the different binning methods to each other, we find that they all perform quite similarly. It would be interesting to further study these and other binning methods for post-hoc canonical calibration.

## H  PROOFS

Proofs appear in separate subsections, in the same order as the corresponding results appear in the paper and Appendix. Proposition 4 was stated informally, so we state it formally as well.

### H.1  STATEMENT AND PROOF OF PROPOSITION 4

**Proposition 4.** *For any predictor $(c, h)$, conf-ECE$(c, h) \leqslant$ TL-ECE$(c, h)$.*

*Proof.* To avoid confusion between the the conditioning operator and the absolute value operator $|\cdot|$, we use $\text{abs}\left(\cdot\right)$ to denote absolute values below. Note that,

$$
\begin{aligned}
\text{abs}\left(P(Y = c(X) \mid h(X)) - h(X)\right) &= \text{abs}\left(\mathbb{E}\left[\mathbb{1}\left\{Y = c(X)\right\} \mid h(X)\right] - h(X)\right) \\
&= \text{abs}\left(\mathbb{E}\left[\mathbb{1}\left\{Y = c(X)\right\} - h(X) \mid h(X)\right]\right) \\
&= \text{abs}\left(\mathbb{E}\left[\mathbb{E}\left[\mathbb{1}\left\{Y = c(X)\right\} - h(X) \mid h(X), c(X)\right] \mid h(X)\right]\right) \\
&\leqslant \mathbb{E}\left[\text{abs}\left(\mathbb{E}\left[\mathbb{1}\left\{Y = c(X)\right\} - h(X) \mid h(X), c(X)\right]\right) \mid h(X)\right] \\
&\qquad \text{(by Jensen's inequality)} \\
&= \mathbb{E}\left[\text{abs}\left(P(Y = c(X) \mid h(X), c(X)) - h(X)\right) \mid h(X)\right].
\end{aligned}
$$

Thus,

$$
\begin{aligned}
\text{conf-ECE}(c, h) &= \mathbb{E}\left[\text{abs}\left(P(Y = c(X) \mid h(X)) - h(X)\right)\right] \\
&\leqslant \mathbb{E}\left[\mathbb{E}\left[\text{abs}\left(P(Y = c(X) \mid h(X), c(X)) - h(X)\right) \mid h(X)\right]\right] \\
&= \mathbb{E}\left[\text{abs}\left(P(Y = c(X) \mid h(X), c(X)) - h(X)\right)\right] \\
&= \text{TL-ECE}(c, h).
\end{aligned}
$$

$\square$

### H.2  PROOF OF THEOREM 1

The proof strategy is as follows. First, we use the bound of Gupta and Ramdas (2021, Theorem 4) (henceforth called the GR21 bound), derived in the binary calibration setting, to conclude marginal, conditional, and ECE guarantees for each $h_l$. Then, we show that the binary guarantees for the individual $h_l$'s leads to a top-label guarantee for the overall predictor $(c, h)$.

Consider any $l \in [L]$. Let $P_l$ denote the conditional distribution of $(X, \mathbb{1}\left\{Y = l\right\})$ given $c(X) = l$. Clearly, $D_l$ is a set of $n_l$ i.i.d. samples from $P_l$, and $h_l$ is learning a binary calibrator with respect to $P_l$ using binary histogram binning. The number of data-points is $n_l$ and the number of bins is $B_l = \lfloor n_l/k \rfloor$ bins. We now apply the GR21 bounds on $h_l$. First, we verify that the condition they require is satisfied:
$$n_l \geqslant k \lfloor n_l/k \rfloor \geqslant 2B_l.$$

Thus their marginal calibration bound for $h_l$ gives,

$$P\left(|P(Y = l \mid c(X) = l, h_l(X)) - h_l(X)| \leqslant \delta + \sqrt{\frac{\log(2/\alpha)}{2(\lfloor n_l/B_l \rfloor - 1)}} \mid c(X) = l\right) \geqslant 1 - \alpha.$$

Note that since $\lfloor n_l/B_l \rfloor = \lfloor n_l / \lfloor n_l/k \rfloor \rfloor \geqslant k$,

$$\varepsilon_1 = \delta + \sqrt{\frac{\log(2/\alpha)}{2(k-1)}} \geqslant \delta + \sqrt{\frac{\log(2/\alpha)}{2(\lfloor n_l/B_l \rfloor - 1)}}.$$

Thus we have

$$P\left(|P(Y = l \mid c(X) = l, h_l(X)) - h_l(X)| \leqslant \varepsilon_1 \mid c(X) = l\right) \geqslant 1 - \alpha.$$

This is satisfied for every $l$. Using the law of total probability gives us the top-label marginal calibration guarantee for $(c, h)$:

$$P(|P(Y = c(X) \mid c(X), h(X)) - h(X)| \leqslant \varepsilon_1)$$

$$= \sum_{l=1}^{L} P(c(X) = l)P(|P(Y = c(X) \mid c(X), h(X)) - h(X)| \leqslant \varepsilon_1 \mid c(X) = l)$$

(law of total probability)

$$= \sum_{l=1}^{L} P(c(X) = l)P(|P(Y = l \mid c(X) = l, h_l(X)) - h_l(X)| \leqslant \varepsilon_1 \mid c(X) = l)$$

(by construction, if $c(x) = l$, $h(x) = h_l(x)$)

$$\geqslant \sum_{l=1}^{L} P(c(X) = l)(1 - \alpha)$$

$$= 1 - \alpha.$$

Similarly, the in-expectation ECE bound of GR21, for $p = 1$, gives for every $l$,

$$\mathbb{E}\left|P(Y = l \mid c(X) = l, h_l(X)) - h_l(X) \mid c(X) = l\right| \leqslant \sqrt{B_l/2n_l} + \delta$$

$$= \sqrt{\lfloor n_l/k \rfloor / 2n_l} + \delta$$

$$\leqslant \sqrt{1/2k} + \delta.$$

Thus,

$$\mathbb{E}|P(Y = c(X) \mid c(X), h_l(X)) - h(X)|$$

$$= \sum_{l=1}^{L} P(c(X) = l)\mathbb{E}|P(Y = l \mid c(X) = l, h_l(X)) - h_l(X)| \mid c(X) = l$$

$$\leqslant \sum_{l=1}^{L} P(c(X) = l)(\sqrt{1/2k} + \delta)$$

$$= \sqrt{1/2k} + \delta.$$

Next, we show the top-label conditional calibration bound. Let $B = \sum_{l=1}^{L} B_l$ and $\alpha_l = \alpha B_l/B$. Note that $B \leqslant \sum_{l=1}^{L} n_l/k = n/k$. The binary conditional calibration bound of GR21 gives

$$P\left(\forall r \in \text{Range}(h_l), |P(Y = l \mid c(X) = l, h_l(X) = r) - r| \leqslant \delta + \sqrt{\frac{\log(2B_l/\alpha_l)}{2(\lfloor n_l/B_l \rfloor - 1)}} \mid c(X) = l\right)$$

$$\geqslant 1 - \alpha_l.$$

Note that

$$\sqrt{\frac{\log(2B_l/\alpha_l)}{2(\lfloor n_l/B_l \rfloor - 1)}} = \sqrt{\frac{\log(2B/\alpha)}{2(\lfloor n_l/B_l \rfloor - 1)}}$$

$$\leqslant \sqrt{\frac{\log(2n/k\alpha)}{2(\lfloor n_l/B_l \rfloor - 1)}} \qquad\qquad \text{(since } B \leqslant n/k\text{)}$$

$$\leqslant \sqrt{\frac{\log(2n/k\alpha)}{2(k-1)}} \qquad\qquad \text{(since } k \leqslant \lfloor n_l/B_l \rfloor\text{)}.$$

Thus for every $l \in [L]$,

$$P(\forall r \in \text{Range}(h_l), \; |P(Y = l \mid c(X) = l, h_l(X) = r) - r| \leqslant \varepsilon_2) \geqslant 1 - \alpha_l.$$

By construction of $h$, conditioning on $c(X) = l$ and $h_l(X) = r$ is the same as conditioning on $c(X) = l$ and $h(X) = r$. Taking a union bound over all $L$ gives

$$P(\forall l \in [L], r \in \text{Range}(h), |P(Y = c(X) \mid c(X) = l, h(X) = r) - r|) \leqslant \varepsilon_2)$$

$$\geqslant 1 - \sum_{l=1}^{L} \alpha_l = 1 - \alpha,$$

proving the conditional calibration result. Finally, note that if for every $l \in [L], r \in \text{Range}(h)$,

$$|P(Y = c(X) \mid c(X) = l, h(X) = r) - r| \leqslant \varepsilon_2,$$

then also

$$\text{TL-ECE}(c, h) = \mathbb{E}|P(Y = c(X) \mid h(X), c(X)) - h(X)| \leqslant \varepsilon_2.$$

This proves the high-probability bound for the TL-ECE. $\qquad\square$

**Remark 3.** Gupta and Ramdas (2021) proved a more general result for general $\ell_p$-ECE bounds. Similar results can also be derived for the suitably defined $\ell_p$-TL-ECE. Additionally, it can be shown that with probability $1 - \alpha$, the TL-MCE of $(c, h)$ is bounded by $\varepsilon_2$. (TL-MCE is defined in Appendix E, equation (17).)

### H.3 PROOF OF PROPOSITION 1

Consider a specific $l \in [L]$. We use $h_l$ to denote the $l$-th component function of $\mathbf{h}$ and $Y_l = \mathbb{1}\{Y = l\}$. Canonical calibration implies

$$P(Y = l \mid \mathbf{h}(X)) = \mathbb{E}\left[Y_l \mid \mathbf{h}(X)\right] = h_l(X). \tag{23}$$

We can then use the law of iterated expectations (or tower rule) to get the final result:

$$\begin{aligned}
\mathbb{E}\left[Y_l \mid h_l(X)\right] &= \mathbb{E}\left[\mathbb{E}\left[Y_l \mid \mathbf{h}(X)\right] \mid h_l(X)\right] \\
&= \mathbb{E}\left[h_l(X) \mid h_l(X)\right] \qquad \text{(by the canonical calibration property (23))} \\
&= h_l(X).
\end{aligned}$$

$\qquad\square$

### H.4 PROOF OF THEOREM 2

We use the bounds of Gupta and Ramdas (2021, Theorem 4) (henceforth called the GR21 bounds), derived in the binary calibration setting, to conclude marginal, conditional, and ECE guarantees for each $h_l$. This leads to the class-wise results as well.

Consider any $l \in [L]$. Let $P_l$ denote the distribution of $(X, \mathbb{1}\{Y = l\})$. Clearly, $D_l$ is a set of $n$ i.i.d. samples from $P_l$, and $h_l$ is learning a binary calibrator with respect to $P_l$ using binary histogram binning. The number of data-points is $n$ and the number of bins is $B_l = \lfloor n/k_l \rfloor$ bins. We now apply the GR21 bounds on $h_l$. First, we verify that the condition they require is satisfied:

$$n \geqslant k_l \lfloor n/k_l \rfloor \geqslant 2B_l.$$

Thus the GR21 marginal calibration bound gives that for every $l \in [L]$, $h_l$ satisfies

$$P\left(|P(Y = l \mid h_l(X)) - h_l(X)| \leqslant \delta + \sqrt{\frac{\log(2/\alpha_l)}{2(\lfloor n/B_l \rfloor - 1)}}\right) \geqslant 1 - \alpha_l.$$

The class-wise marginal calibration bound of Theorem 2 follows since $\lfloor n/B_l \rfloor = \lfloor n/\lfloor n/k_l \rfloor \rfloor \geqslant k_l$, and so

$$\varepsilon_l^{(1)} \geqslant \delta + \sqrt{\frac{\log(2/\alpha_l)}{2(\lfloor n/B_l \rfloor - 1)}}.$$

Next, the GR21 conditional calibration bound gives for every $l \in [L]$, $h_l$ satisfies

$$P\left( \forall r \in \mathrm{Range}(h_l), |P(Y = l \mid h_l(X) = r) - r| \leqslant \delta + \sqrt{\frac{\log(2B_l/\alpha_l)}{2(\lfloor n/B_l \rfloor - 1)}} \right) \geqslant 1 - \alpha_l.$$

The class-wise marginal calibration bound of Theorem 2 follows since $B_l = \lfloor n/k_l \rfloor \leqslant n/k_l$ and $\lfloor n/B_l \rfloor = \lfloor n/\lfloor n/k_l \rfloor \rfloor \geqslant k_l$, and so

$$\varepsilon_l^{(2)} \geqslant \delta + \sqrt{\frac{\log(2B_l/\alpha_l)}{2(\lfloor n/B_l \rfloor - 1)}}.$$

Let $k = \min_{l \in [L]} k_l$. The in-expectation ECE bound of GR21, for $p = 1$, gives for every $l$,

$$\mathbb{E}\left[\text{binary-ECE-for-class-}l\ (h_l)\right] \leqslant \sqrt{B_l/2n_l} + \delta$$
$$= \sqrt{\lfloor n/k_l \rfloor /2n} + \delta$$
$$\leqslant \sqrt{1/2k_l} + \delta$$
$$\leqslant \sqrt{1/2k} + \delta.$$

Thus,

$$\mathbb{E}\left[\text{CW-ECE}(c, h)\right] = \mathbb{E}\left[ L^{-1} \sum_{l=1}^{L} \text{binary-ECE-for-class-}l\ (h_l) \right]$$
$$\leqslant L^{-1} \sum_{l=1}^{L} (\sqrt{1/2k} + \delta)$$
$$= \sqrt{1/2k} + \delta,$$

as required for the in-expectation CW-ECE bound of Theorem 2. Finally, for the high probability CW-ECE bound, let $\varepsilon = \max_{l \in [L]} \varepsilon_l^{(2)}$ and $\alpha = \sum_{l=1}^{L} \alpha_l$. By taking a union bound over the the conditional calibration bounds for each $h_l$, we have, with probability $1 - \alpha$, for every $l \in [L]$ and $r \in \mathrm{Range}(h)$,

$$|P(Y = l \mid c(X) = l, h(X) = r) - r| \leqslant \varepsilon_l^{(2)} \leqslant \varepsilon.$$

Thus, with probability $1 - \alpha$,

$$\text{CW-ECE}(c, h) = L^{-1} \sum_{l=1}^{L} \mathbb{E}|P(Y = l \mid h_l(X)) - h_l(X)| \leqslant \varepsilon.$$

This proves the high-probability bound for the CW-ECE. $\qquad\square$

