# OpenReview forum: "Top-label calibration and multiclass-to-binary reductions"
_ICLR.cc/2022/Conference — ICLR 2022 Poster_

### Official Review · Reviewer_NPUy · 2021-10-29

**Correctness:** 3
**Technical Novelty And Significance:** 4
**Empirical Novelty And Significance:** 3
**Recommendation:** 6
**Confidence:** 3

**Main Review:**

This is a pretty dense paper that contains a lot of material, but it is very well written. I do not consider myself an expert on the topic, but I could follow the flow of the paper quite well.

The authors have been able to convince me of the disadvantages of confidence calibration, and the advantages of the method they introduce. To this end, Example 1 and the case study in Figure 1 really helped. The experimental results also seem to support the claims of the authors. From a more conceptual perspective, the proposed algorithms are also appealing.

However, I do see potential problems with the approach of the authors for classification problems with infrequent classes, like in extreme multi-class classification, where long-tail classes have very few observations. In such situations, confidence calibration will probably work much better, as one doesn't have to condition for rare classes. Conversely, for the approach of the authors, one needs much more observations per class. So, in this regard, the experiments are currently somewhat limited, and probably telling a too optimistic story. I would have liked to see some experiments with extreme classification datasets that have long-tail class distributions.

**Summary Of The Paper:**

This paper discusses the topic of classifier calibration in multi-class classification. The focus is on obtaining well-calibrated probabilities for the top-predicted classes. The authors give arguments, examples and experimental results to show that the commonly-used confidence calibration method suffers a number of shortcomings that makes it less useful in practice. As an alternative, top-label calibration is proposed, where calibration is analyzed on a per-class basis, when a specific class is the top class.

In addition, the authors discuss shortcomings of confidence reliability diagrams, and they propose multi-class to binary reductions for achieving top-label calibration. In the experiments several methods are compared on two classical image classification benchmarks.

**Summary Of The Review:**

Interesting and well-motivated idea. Datasets in the experiments are quite simple.

---

> ### Author Response · Authors · 2021-11-19
> **Experiment with top-label calibration and tail classes**
>
> Thank you for your positive reception of our work. We fully agree that exploring top-label calibration for extreme multi-class classification is interesting avenue for future work. While we did not have an extreme multi-class dataset, we did in fact have a case study with a class-imbalanced dataset, in which we show that TL-HB adapts to rare tail classes. Due to space constraints, this case study had to be moved to Appendix B.2.
>
> ---
>
> **Summary of the experiment in Appendix B.2**: We explore post-hoc calibration of random forests with the class-imbalanced COVTPE-7 dataset, for which the most likely class of the 7 classes is roughly 100 times more likely than the least likely class. The way the top-label calibrator works, the binary calibration for each class is carried out independently, thus automatically enabling low-occurence classes to be treated differently from high-occurence classes. In the case of top-label-HB, we show that a different number of bins per class can be used based on the number of times the given class is predicted as the top-label. In our experiment, for the least likely class, top-label-HB automatically used a single bin, whereas for the most likely class, top-label-HB used about 200 bins. This adaptivity leads to a significant improvement in the overall top-label calibration performance as illustrated in Figure 4.
>
> ---
>
> _Changes to the paper:_ This experiment was there in the previous paper and referenced in Section 4, but mentioned somewhat in passing. In the revised version, we now point the reader to this experiment more explicitly; see text in blue at the end of the 'Experimental details section'. Although we found this experiment very interesting and insightful, we were unable to move it to the main paper due to space constraints.

---

> > ### Comment · Reviewer_NPUy · 2021-11-30
> > **Response to previous comment**
> >
> > Thanks for the response. There are quite some extreme multi-class datasets that could be used, see e.g. Imagenet or the ones used in Mortier et al. "Efficient set-valued prediction in multi-class classification" Data Mining and Knowledge Discovery 2021. However, I do not see this as an absolute requirement to have the paper accepted.

---

### Official Review · Reviewer_xjQt · 2021-11-01

**Correctness:** 3
**Technical Novelty And Significance:** 2
**Empirical Novelty And Significance:** 2
**Recommendation:** 5
**Confidence:** 3

**Main Review:**

The first seven pages of the paper are very nice, and I very much enjoyed reading this material. The problems start with the experiments:

1) Reading through the appendix, the authors are obviously aware that a calibration set is normally used to calibrate a model, before it is evaluated on the test set to compute ECE. However, there is no reference to how this is done for the experiments in the paper, which are based on pre-trained models. It seems that the validation set was perhaps used for both building the discretization-based calibration models and their evaluation, which seems highly problematic.

2) One of the three empirical observations on Page 9 states that the new TL-HB is the best performing method for 1-vs-rest ECE. In fact, it is the un-normalized variant of 1-vs-rest calibration that is shown in Table 3 and that performs best! Assuming this is just a typo and "TL-HB" in Observation (c) was meant to be "CW-HB", this is the most surprising and potentially most impactful finding in the paper. (The other two observations are about the behaviour wrt TL-ECE/MCE, which are the new, less obvious ECE metrics, and the results are also more mixed wrt these metrics.) However, there is no analysis at all in the paper why leaving out normalization in the 1-vs rest method (CW-HB) is so much better than using it (N-EB)!

3) There is no comparison to isotonic regression, which is trivially applied using the 1-vs-rest method, and like binning, is also a non-parameteric method. The scaling-based methods are all parameteric.

4) The number of datasets is very limited (CIFAR-10 and CIFAR-100).

5) The actual 1-vs-rest discretization-based calibration algorithm used in the paper (CW-HB) seems non-standard because a separate binning is applied for each class (see Algorithm 9 in Appendix C.1, assuming that Binary-histogram-binning performs equal-frequency binning as indicated elsewhere in the paper). My understanding is that normally, the same bin boundaries are used for all classes when this method is applied. The effect of this is unclear. Could it explain why normalization performs so poorly in Table 3?

The submission describes the discretization-based algorithms being evaluated as special instances of a general M2B "notion" for calibration of multi-class problems, and there is a corresponding "general-purpose" calibration algorithm that includes the evaluated algorithms as special cases. It is unclear how helpful this algorithm and the M2B "notion" are. This aspect of the paper seems somewhat trivial. There are no results or proofs regarding this general-purpose formulation of the algorithm.

Longer and longer appendices seem to be becoming the norm, but this submission is quite extreme in this regard, particularly because only some appendices are referred to in the main text (G, A, D.3, D.2, D.4, D.5, B, D.1) while others aren't: E (random forest experiments, extend Appendix B.2), C (CW-HB), F (canonical multi-class calibration, 8 pages). The role of appendix F in particular is mysterious: it is only tangentially related to what is presented in the main text.

Other questions and comments:

- Do the calibrated probabilities obtained using top-label calibration sum to 1?

- "However, the distribution of $g$ can be different for different labels, thus they should be treated differently" -- I don't understand the reason for having this sentence here. Is this to reinforce that point? I would delete it.

- Renumber the algorithms to follow the order in which they are discussed.

- "the expectation is over the calibration data" - isn't it trivial to fit the calibration data arbitrarily well? Why is this bound useful then?

- Is N-HB defined in the text or just in the caption?

- Why is CW-HB not included in Table 2? Is it obvious that it performs worse than TL-HB for these metrics? Conversely, doesn't it make sense to include TL-HB in Table 3?

**Summary Of The Paper:**

The paper proposes a new method to fix confidence calibration by conditioning on the predicted label when formulating the calibration task (as well as on the class probability estimate for that label, as done in confidence calibration). This new approach is called "top-label" calibration. The paper shows how to modify the calibration process to achieve this by building a separate calibration model for each predicted class value. Experiments compare this approach, using binning as the calibration method, to confidence calibration as well as standard 1-vs-rest calibration (both with and without normalization in the latter method) based on calibration of pre-trained CIFAR-10/100 CNNs. Temperature scaling, vector scaling, and Dirichlet scaling are also included. Evaluation is performed using 1-vs-rest ECE, a new top-label ECE, and a maximum calibration error (MCE) variant of the latter. Surprisingly, the proposed approach does not perform as well as 1-vs-rest calibration when using the top-label variant of ECE; it does perform a lot better wrt the MCE variant on the CIFAR-10 data though. When considering 1-vs-rest ECE, the paper reports that 1-vs-rest *without* normalization performs better than the same approach with normalization, and also better than all scaling-based methods. The paper also explains how to adapt top-K-confidence calibration to obtain top-K-label calibration, but this is not evaluated.

**Summary Of The Review:**

There are several problems with the empirical evaluation and the analysis. The submission (including the many appendices) also seems heavily overloaded, to the point that the main message becomes quite unclear.

---

> ### Author Response · Authors · 2021-11-19
> **Key clarification: all empirical results are on unseen test data, theory is also for unseen test data (Response part 1 of 3)**
>
> Thank you for the detailed comments. We wish to clarify that **all evaluations were done on unseen test data**, not the validation data. We apologize for this misunderstanding and have revised the paper to make this clearer. Specific responses to your concerns and a summary of the changes to the paper, are below.
>
> You mentioned
>
> > It seems that the validation set was perhaps used for both building the discretization-based calibration models and their evaluation, which seems highly problematic.
>
> We used the standard train-validation-test split of 45000/5000/10000 images for both CIFAR-10 and CIFAR-100. Our splits were in fact _identical_ to that of Mukhoti et al. [2020], and our CW-ECE numbers for the base and base+temp_scaling models match exactly with theirs (compare the first two columns of our Table 2 with the 3rd and 4th columns of Table F.2 of Mukhoti et al. [2020]).
>
> Although we mentioned in Section 2.1: "Figure 1 illustrates the (test-time) calibration performance of a ResNet-50 model...", we agree that this information needs to be made clearer and easier to find.
>
> _Change to the paper_: In the "Experimental details" paragraph on page 9, we have now clearly mentioned the split sizes as above and the fact that all results are on test data.
>
> Additionally, you asked
>
> > "the expectation is over the calibration data" - isn't it trivial to fit the calibration data arbitrarily well? Why is this bound useful then?
>
> Again, we clarify that the theory is in fact over test data. TL-ECE and CW-ECE are defined as the expected calibration error for an **independent test-point** $(X, Y)$ (equations (4) and (9)). The expectation over the calibration data corresponds to the randomness in the predictor which is learnt on random calibration data. Thus the bound should be read as
> $$
> E_{\text{predictor}}[\text{test-ECE(predictor)}] \leq \sqrt{1/2k}.
> $$
>
> _Change to the paper_: We have paraphrased the above in the paper immediately after the theoretical result is mentioned.
>
> In light of these serious misunderstandings, we hope that you will reconsider your score. Our responses to other concerns (not surrounding this misunderstanding) are in the next comment.

---

> > ### Author Response · Authors · 2021-11-19
> > **Responses to other major concerns and questions + corresponding updates to the paper (Response part 2 of 3)**
> >
> > In the reviewer summary, you wrote:
> >
> > > The submission (including the many appendices) also seems heavily overloaded, to the point that the main message becomes quite unclear.
> >
> > We have now revised the paper to make the main messages clearer, among other things: points (b) and (c) in the introduction have been revised, and all Appendices have been referenced appropriately. You mentioned:
> >
> > > The role of appendix F in particular is mysterious: it is only tangentially related to what is presented in the main text.
> >
> > We felt that understanding canonical calibration was crucial before diving into other notions of multiclass calibration, and hence performed this study. Based on this study, we concluded that canonical calibration needs exponential sample complexity, hence leading to the focus on M2B notions. Due to the limited practical applicability of canonical calibration, we moved this study to the Appendix.
> >
> > Regarding the non-normalized-HB you mentioned,
> >
> > > ... Assuming this is just a typo and "TL-HB" in Observation (c) was meant to be "CW-HB"... CW-HB is better than N-HB: this is the most surprising and potentially most impactful finding in the paper... why leaving out normalization is so much better?
> >
> > Yes, the TL-HB is a typo: CW-HB does the best for both datasets and all architectures for CW-ECE.
> >
> > It turns out that we can show calibration guarantees for CW-HB --- the theorem is in Appendix C (as indicated in the *Formal algorithm and theoretical guarantees* paragraph on page 9). On the other hand, it is unclear how to show guarantees for normalized-HB (N-HB). Ours is one of the first papers to report that not normalizing helps for CW-ECE, and we provided one plausible theoretical reasoning for this in the distribution-free framework. There is definitely more to be said, and we expect future works to explore this further.
> >
> > _Change to the paper_: In point (c) of the findings, we now paraphrase the above point.
> >
> > > no comparison to isotonic regression
> >
> > Given the number of binary and multiclass calibrators that exist in literature, we had to restrict experimentation in some way. In our early experiments, we observed that isotonic regression makes bins with many calibration points, due to which the calibration becomes somewhat trivial (compared to histogram binning where the mass is uniform by construction). This observation was also made by Gupta and Ramdas [2021, Section 4]. We thus removed isotonic regression from consideration for the final experiments section.
> >
> > > The actual 1-vs-rest discretization-based calibration algorithm used in the paper (CW-HB) seems non-standard because a separate binning is applied for each class...
> >
> > To the best of our knowledge, the standard application of HB to multiclass corresponds to solving $L$ different 1-v-all problems without any sharing of information such as the bin boundaries (as is done by CW-HB and N-HB). At least, this is the implementation in the works of Zadrozny and Elkan [2002], Guo et al. [2017] and Kull et al. [2019], where sharing of bin boundaries is not mentioned.
> >
> > You raised the following questions regarding the utility of M2B calibration:
> >
> > > It is unclear how helpful this algorithm and the M2B "notion" are.
> >
> > The following line (slightly paraphrased) that has been added at the beginning of Section 3.2:
> >
> > > An immediate consequence of the M2B reduction is that one can now solve multiclass calibration problems by leveraging the well-developed methodology for binary calibration.
> >
> > Thus M2B calibration allows us to treat many notions of calibration in a unified manner using reductions to the (much more well studied) binary case. Indeed, if tomorrow someone comes up with another notion of calibration that is more suitable to (say) a structured classification problem, it is likely that our M2B reductions will come to their aid in developing calibrators to achieve the new notions.
> >
> > Empirically, we showed that two completely new calibration algorithms that fall out of the M2B framework, CW-HB and TL-HB, are in some cases the best-performing methods compared to many previous multiclass calibrators. Further, if one is interested in, say, top-K-label calibration, then top-K-label-HB (= Algorithm 6 + binary-HB) can simply be implemented 'out-of-the-box'. This is a completely new algorithm, just like TL-HB and CW-HB, but tuned to top-K-label-ECE. You mentioned that,
> >
> > > There are no results or proofs regarding this general-purpose formulation of the algorithm.
> >
> > While we have shown guarantees explicitly only for TL-HB and CW-HB, guarantees can also be shown for any other M2B calibrator with HB, following the same proof technique as Theorems 1 and 2.

---

> > > ### Author Response · Authors · 2021-11-19
> > > **Responses to other minor comments (Response part 3 of 3)**
> > >
> > > The responses to the six questions in 'other questions and comments' are in order below:
> > >
> > > - For a single test-point $x$, the top-label calibrator gives only a single confidence score $h(x) \in [0,1]$ corresponding to $c(x)$, and not a probability vector on the simplex.
> > > - This sentence has been removed.
> > > - Done.
> > > - This is discussed in the Response part 1 of 3.
> > > - It is now described in text as well, in the 'Formal algorithm and theoretical guarantees' part.
> > > - Our goal was to compare the calibrator that is directly instantiated from the general-purpose M2B calibrator, using the corresponding M2B notion. Thus for TL-ECE we used TL-HB and for CW-ECE we used CW-HB.

---

> > > ### Comment · Reviewer_xjQt · 2021-11-22
> > > **Other major concerns**
> > >
> > > The lack of a theoretical guarantee for normalized-HB does not explain why it performs so much worse (for class-wise ECE). The theory provided does *not* indicate that skipping normalization improves performance: it simply shows that performance guarantees can be obtained if normalization is not performed.
> > >
> > > I could not follow the justification for not including isotonic regression in the experiments, even after consulting the Gupta and Ramdas [2021] reference provided: surely, performance is what matters, not whether the calibration "becomes somewhat trivial".
> > >
> > > I do concede that my claim regarding the unusualness of per-class bin boundaries seems to be incorrect, at least for the equal-frequency-based discretization method applied in the submission: although the descriptions in Zadrozny and Elkan [2002] and Guo et al. [2017] seem ambiguous in this regard, the publicly available code by Kull et al. [2019] uses per-class bin boundaries. Of course, if *equal-width* discretization is applied, the bin boundaries will be consistent across classes. Interestingly, the results in Gupta and Ramdas [2021] show that equal-width can perform well in some regimes, so some discussion of why this method is not evaluated in the submission seems warranted.

---

> > > > ### Author Response · Authors · 2021-11-29
> > > > **Re:**
> > > >
> > > > We agree that using HB is a subjective choice since isotonic regression or fixed-width binning may give better performance depending on the notion of ECE/MCE. However, the main message of the paper is not about HB vs other binary calibrators, but proposing a single agnostic framework for achieving multiple notions of multiclass calibration using _any_ binary calibrator.
> > > >
> > > > Thus, our experiments are illustrative, not comparative, and our choice to focus on HB is due to its distribution-free guarantees, as well as the fact that previous papers have consistently reported poor numbers for HB on multiclass calibration tasks (since the wrong multiclass-to-binary reduction was used). The reductions, and the notion of top-label calibration, are conceptual advances that are not tied to any one binary calibrator.

---

> > ### Comment · Reviewer_xjQt · 2021-11-22
> > **Clarification acknowledged**
> >
> > Thank you for clarifying the protocol used for splitting the data. I have raised my score.

---

### Official Review · Reviewer_ZncC · 2021-11-02

**Correctness:** 4
**Technical Novelty And Significance:** 3
**Empirical Novelty And Significance:** 3
**Recommendation:** 8
**Confidence:** 4

**Main Review:**

This paper is very well written, tackles an interesting area, proposes a novel calibration, and demonstrates the utility in empirical experiments.

Strengths:

- The proposed calibration results in a more intuitive interpretation than previous approaches and the authors argue so convincingly.
- Matching calibration algorithms and metrics places everything into a nice framework, and the experiments demonstrate clearly the value of matching the algrithm and metric.
- Histogram binning, within the appropriate algorithm, is shown to be a competitive post-hoc calibration technique.

Weaknesses:

- Only histogram binning is considered, though there are other post-hoc calibration methods that clearly fit in the algorithms described. It would have been good to see some results, or at least a discussion.

**Summary Of The Paper:**

The authors consider the problem of calibrated probabilistic outputs for multiclass classifiers. They consider the various types of calibrations previously studied, such as confidence calibration and class-wise calibration, and then propose a novel notion of calibration based on the choice of top-labels.

**Summary Of The Review:**

Excellent paper with good results and clear practical applications. The work is also well placed within existing literature.

---

> ### Author Response · Authors · 2021-11-19
> **Response to reviewer ZncC**
>
> Thank you for your positive reception of our work.
>
> We focused on histogram binning since it is the only binary calibrator that we know of with both (i) distribution-free calibration guarantees, and (ii) strong empirical performance on binary calibration datasets. We briefly allude to these reasons in the first two paragraphs of Section 4. Purely methodologically speaking (if one does not care about guarantees), one can plug in any binary calibrator into the generic reduction schemes, and it would be interesting to see the empirical results.

---

### Official Review · Reviewer_8zku · 2021-11-03

**Correctness:** 3
**Technical Novelty And Significance:** 3
**Empirical Novelty And Significance:** 2
**Recommendation:** 5
**Confidence:** 4

**Main Review:**

Strengths: The paper makes a convincing argument that on an individual level, confidence calibration may be misguided while their definition makes more sense. It is also true that a natural algorithm to achieve many notions of multi-label calibration is to reduce it to the binary case, as the paper suggests.
Weaknesses: I think the definition also has obvious drawbacks that the authors do not discuss. In particular I am not convinced that calibrating only the top label makes sense. In practice it just means that you partition the data by the top label and then calibrate each partition separately (as the algorithm they suggest effectively does). The predictions outside the top label make no difference. It should be noted that satisfying this requirement is very easy. Pick the most common label and assign to all points the expectation of that label. Thus, the point of calibration is to do it to an existing classifier *with out* sacrificing other good properties such as loss minimization. Since the top label doesn't change by the calibrator the accuracy is unchanged, but the typical loss function for multi-class is cross entropy.

Detailed comments:
- Please explain better why calibrating only the top label is sufficient. Typically we assume that c(x) returns a vector of probabilities of dimension L, not just the top label out of the L.
- At the beginning of section 2, please define confidence better. The arg max of the expression is a class (a number in [L]), not a pair (c,h), so the definition is confusing.
- I am not very familiar with confidence calibration as is defined here, but it must have some good properties no? Please discuss them and contrast with your definition as well.
- I'm not sure I fully understand the contribution of Section 3. Sure, some notions are reductions from binary classifiers, so lend themselves to be computed via binary calibrators. Is there anything more you can say? (for instance, is error being compounded?, is loss minimization being affected? are there computational tradeoffs?).
- In table 2 and 3 it is worth noting that scaling algorithms are not designed to bring down ECE or TL ECE.

**Summary Of The Paper:**

The paper suggests a definition for calibration in the multi-class setting named 'top label calibration'. The idea is to have only the most likely class calibrated. The small difference with previous definitions is that instead of conditioning only only the confidence value, the conditioning here is also on the identity of the class. The authors argue convincingly that this conditioning renders the definition more meaningful.
The authors then observe that many definitions for multi-class calibration could be reduced to multiple instances of binary calibration and suggest an algorithmic framework where a binary calibrator is used as a black box to achieve the multi class calibrator. They then test this by instantiating it with histogram binning and measuring the corresponding notion of expected calibration error.

**Summary Of The Review:**

The provides a variation over  confidence calibration that takes into account the identity of the top label. This variation makes sense under some scenarios, but is still a quite a weak notion on its own. While it's an interesting notion to discuss, I think the contributions of the paper are too thin to justify publications.

---

> ### Author Response · Authors · 2021-11-19
> **Response to questions around top-label calibration + corresponding updates to the paper (Response 1 of 2)**
>
> Thank you for your review. We have responded to your questions below, and wherever feasible, revised the paper to reflect these answers.
>
> First, we focus on the following question from your review:
>
> > Please explain better why calibrating only the top label is sufficient.
>
>
> Calibrating the top label may not always be sufficient. In some cases, it may make more sense to calibrate the top 2 or 3 (or more) labels instead, as mentioned in Section 3.2 (top-$k$-label calibration). Or one may choose to calibrate the entire prediction vector, as discussed in Appendix F (canonical calibration). However, there are computational and statistical tradeoffs. For example, the binning of a $k$-dimensional probability simplex requires exponentially more bins (in $k$) and thus exponentially more samples as well to achieve a reasonable sharpness and calibration (one would need to avoid empty bins, etc). The final choice of which notion to choose depends on the domain requirements and the amount of data available.
>
>
> We highlight top-($k=1$)-calibration because if the classifier has high accuracy, calibrating just the top label could be sufficient for interpretability and usability of such models. For instance, if a high-accuracy cancer-grade-prediction model predicts a patient as having "95\% grade I, 3\% grade II, and 2\% grade III", we would suggest the patient to undergo an intrusive treatment. However, we may want to know (and control) the number of non-grade-III patients that were given this suggestion incorrectly. In other words, we may want to know: is Prob(cancer is not grade III $\mid$ cancer is predicted to be of grade III with confidence 95%) = 5%? Top-label calibration captures this intuition.
>
> (The above example has been added to the paper in a slightly different context; see blue text on page 2.)
>
> > I am not very familiar with confidence calibration as is defined here, but it must have some good properties no?
>
> Both top-label calibration and confidence calibration have the same motivation. They are 'low requirement' calibration notions that are most applicable if the base model has high accuracy. Since this is commonly true for deep learning models, confidence calibration has had a significant influence on deep model calibration. However, as we illustrate in the paper, the calibration guarantee given by confidence calibration does not have a practically meaningful interpretation.
>
> You mentioned:
>
> > It should be noted that satisfying top-label calibration is very easy. Pick the most common label and assign to all points the expectation of that label.
>
> Such drawbacks apply to every notion of calibration. You suggested two 'hacks' to achieve top-label calibration easily:
>
> 1. One can predict a single label (such as the most frequent top predicted class, along with its frequency) for every point $x$. But this can also be done for confidence calibration, where a single label can be predicted for every $x$. And for every other similar notion of calibration.
>
> 2. A single probabilistic prediction can be made for all points $x$. This can also be done for binary calibration. Suppose $Y \in \\{0,1\\} $ and $P(Y=1) = 0.5$, then $h(x) \equiv 0.5$ is calibrated: $P(Y  = 1\mid h(X) = 0.5) = P(Y=1) = 0.5$. In other words, for balanced binary datasets, predicting 0.5 at every point is calibrated: this is well known, so your criticism is not of our notion of top-label calibration, but of calibration as a metric.
>
> Thus the question is, does the calibration algorithm being used rely on the hacks (1) or (2) above? If it does, it would be a poor calibration algorithm. _However, all calibration algorithms that we propose (or compare to) do not use such hacks._ Specifically,
>
> 1. Our top-label calibrator does not change the existing predicted label $c(x)$ and the accuracy of the base model remains unchanged. (Predicting an identical label at every $x$ would hurt accuracy hugely.)
>
> 2. The histogram binning algorithms we espouse do not return a single probability, but one probability per bin. The results in Tables 2 and 3 use #bins B = 15.
>
> Overall, the points are assigned upto $B \times L$ different probabilities (since the binning is separate per class), and the value of $B$ can be adjusted based on the amount of available calibration data. You also mentioned:
>
> > Since the top label doesn't change, the accuracy is unchanged, but the typical loss function for multi-class is cross-entropy.
>
> Cross-entropy loss is a surrogate loss for misclassification error, that is useful for training high-accuracy deep nets. Once the base deep net with high accuracy is learnt, we ask the post-hoc question: "can the calibration for the top-label be improved without sacrificing the high accuracy?" Top-label-HB does exactly this.
>
> Our response continues in the next comment.

---

> > ### Author Response · Authors · 2021-11-19
> > **Response to questions around M2B framework + corresponding updates to the paper (Response 2 of 2)**
> >
> > Regarding the M2B framework, you asked:
> >
> > >  ... some notions are reductions from binary classifiers, so lend themselves to be computed via binary calibrators. Is there anything more you can say?
> >
> > The observation that most popular notions of multiclass calibration rely on binary calibration statements is not stated explicitly and acknowledged in any generality in other papers on these topics. We make this explicit and then exploit it. This insight enables us to treat many notions of calibration in a unified manner using reductions to the (much more well studied) binary case. Indeed, if tomorrow someone comes up with another notion of calibration that is more suitable to (say) a structured classification problem, it is likely that our M2B reductions will come to their aid in developing calibrators to achieve the new notions. A number of changes have been made in the introduction and the beginning and end of Section 3.2 to help make this point clearer to other readers. In particular, we have added the following summarizing line (slightly paraphrased) at the beginning of Section 3.2:
> >
> > > An immediate consequence of the M2B reduction is that one can now solve multiclass calibration problems by leveraging the well-developed methodology for binary calibration.
> >
> > Thus for any given M2B notion, a novel and customized M2B calibrator can be constructed that is specifically tuned to the given notion. Empirically, we evaluated two completely new calibration algorithms that fall out of the M2B framework: CW-HB and TL-HB. These new algorithms are in some cases the best-performing methods compared to many previous multiclass calibrators. Further, if one is interested in, say, top-K-label calibration, then top-K-label-HB (= Algorithm 6 + binary-HB) can simply be implemented 'out-of-the-box'. This is a completely new algorithm, just like TL-HB and CW-HB, but tuned to top-K-label-ECE. While in the paper we have shown guarantees explicitly only for TL-HB and CW-HB, guarantees can also be shown for any other M2B calibrator with HB, following the same proof technique as Theorems 1 and 2.
> >
> > You also asked the following specific questions regarding M2B caibration:
> >
> > > is error being compounded?
> >
> > We are not sure which error you mean, but generally, the more the number of binary calibration claims that are being made, the harder it would be (in terms of sample-complexity) to satisfy them simultaneously.
> >
> >
> > > is loss minimization being affected?
> >
> > For top-label calibration and misclassification loss, the answer is no.
> >
> > > are there computational tradeoffs?
> >
> > The computational complexity depends on the number of binary calibration claims being made by the corresponding M2B notion.
> >
> >
> > Other suggested changes in the review:
> >
> > > At the beginning of section 2, please define confidence better.
> >
> > This change has been made.
> >
> > > In table 2 and 3 it is worth noting that scaling algorithms are not designed to bring down ECE or TL ECE.
> >
> > We have added a comment regarding this in the main text.

---

> > > ### Comment · Reviewer_8zku · 2021-11-21
> > > **RE**
> > >
> > > I thank the authors for their thoughtful and detailed response.
> > > I agree that the notion of top label calibration is relevant especially in the high accuracy regime as the example you provided demonstrated. It is my opinion that in this case, a binary calibrator of label vs everyone else, for each label, is a natural thing to do. I recognize that it had not been explicitly stated before and I appreciate the authors putting these observations into a nice framework and the experimentation they have done. It is simply my subjective opinion (as a reviewer) that this does not add up to a submission that is strong enough for ICLR, so I would not raise my score. I think the notion of multi-class calibration is still far from understood, and, respectfully, I don't see this paper as making a big enough step towards improving our understanding.

---

> > > > ### Author Response · Authors · 2021-11-24
> > > > **Brief comment**
> > > >
> > > > Thank you for acknowledging our response.
> > > >
> > > > >  It is my opinion that in this case, a binary calibrator of label vs everyone else, for each label, is a natural thing to do.
> > > >
> > > > Just to make sure this has not been missed, we wished to clarify that the top-label calibrator (Algorithm 1) is different from the one-versus-all calibrator (Algorithm 2). Both algorithms are performing statistically different tasks with different operational interpretations.

---

### Author Response · Authors · 2021-11-19
**Paper has been revised based on the received feedback**

Dear reviewers and area chair,

Thank you for the reviews. Based on the comments, we have revised the paper focusing on three key areas:

1. Addition clarifying remarks have been added in the experiments section (Section 4). Specifically, it was earlier unclear that evaluation was indeed performed on unseen test data, as pointed out by Reviewer xjQt. This has been rectified.
2. Points (b) and (c) in the introduction and Section 3 have been improved to make the contribution of M2B calibration clearer.
3. To motivate top-label calibration further, a short example has been added in Section 2.

**To make it easier to review the updates to the paper, major updates have been made in blue text. If the paper is accepted, the blue text will be removed for the camera-ready version.**

---

### Decision · Program_Chairs · 2022-01-20

**Decision:**

Accept (Poster)

**Comment:**

The paper studies the problem of multi-class calibration, proposing new notion of "top-label calibration", and presenting and comparing new algorithms for multi-class calibration. Reviewers generally found the paper to be well-written, and tackling a foundational problem. There were some questions regarding the experiments:

(1) _Lack of explanation of why unnormalised beats normalised._ One of the paper's main empirical findings is that using an unnormalised predictor with histogram binning (CW-HB) can significantly outperform a normalised one (N-HB). There is however limited discussion prior to this of why such behaviour is expected.

(2) _Lack of comparison to isotonic regression._ One can apply isotonic regression in conjunction with one of the M2B algorithms in Sec 3. It is of interest whether isotonic regression + a suitable M2B wrapper compares to HB + an M2B wrapper.

(3) _Comparison to OVA calibration_. One reviewer raised the concern that the new algorithms proposed in this work are not too surprising; e.g., using a binary calibrator of one label vs everyone else appears natural.

(4) _Overloaded appendices_. One reviewer pointed out that some material in the Appendix is not referenced in the body, thus making the work not self-contained.

For point (1), the response indicates that the unnormalised model can obtain distribution-free guarantees. The lack of such a guarantee for the normalised model does not suggest that such a guarantee is impossible, however. Certainly the present work need not solve this issue in entirety, but if this is intended to be a main takeaway of the experiments, a little more discussion in the body seems advisable.

On this note, from my reading, the experiments seek to demonstrate that suitable M2B reductions can dramatically improve the performance of HB. However, with the use of multiple evaluation metrics (Top-ECE, Top-MCE, Classwise-ECE) it is not clear if the authors intend to promote one specific M2B reduction as generally favourable; further, the text in Sec 3.2 suggests that N-HB is the method considered in prior works, which then seems to do better on top-label ECE than prior works. Which suggests that for this particular metric, one does not gain much from other M2B reductions?

For point (2), the response argued that "the main message of the paper is not about HB vs other binary calibrators, but proposing a single agnostic framework for achieving multiple notions of multiclass calibration using any binary calibrator". From my reading of the paper, I think this claim is accurate, and agree that the conceptual advance is the generic M2B framework itself

For point (3), the authors responded to suggest that while Algorithm 2 performs a natural one-versus-all calibration, this is different from Algorithm 1. Further, the latter is shown to be useful in Table 2 (bottom panel).

For point (4), the revision involved referencing relevant material in the Appendix. These seem better, though I would prefer if theorem statements (e.g., Theorem 1) appear in full or as sketches in the body.

Further to the above, I have a couple of minor suggestions:
- consider removing most hline's from Tables 1 -- 3

- keep the ordering of Algorithms 1 -- 4 the same as that in which methods are presented in Table 1.

**Summary**. The paper considers a foundational problem. It makes one simple yet interesting contribution in its definition of top-label calibration. Detailing the various multi-class calibrators in Section 3 is another contribution: albeit simple, it does illuminate the subtle issue of the role of normalisation in post-hoc calibration, which empirically is shown to have non-trivial impact. There are certainly avenues where the paper could be further strengthened, but overall I do see it as potentially being of broad interest to the community, and inspiring future work. My recommendation is thus for the authors to further incorporate the reviewers' detailed suggestions and the comments above, which can broaden the clarity, scope and impact of the work.